# Global analysis of in situ cosmogenic $^{26}$Al and $^{10}$Be and inferred erosion rate ratios in modern fluvial sediments indicates widespread sediment storage and burial during transport

Christopher T. Halsted[1], Paul R. Bierman[2], Alexandru T. Codilean[3], Lee B. Corbett[2], Marc W. Caffee[4]

[1]Department of Geosciences, Williams College, Williamstown, MA 01267, USA
[2]Rubenstein School of Environment and Natural Resources, University of Vermont, Burlington, VT 05405, USA
[3]School of Earth, Atmospheric and Life Sciences and Australian Research Council Centre of Excellence for Australian Biodiversity and Heritage (CABAH), University of Wollongong, Wollongong NSW 2522, Australia
[4]Department of Physics and Astronomy, Purdue University, West Lafayette, IN 47907, USA

*Correspondence to*: Christopher T Halsted (ch22@williams.edu)

**Abstract.** Since the 1990s, analysis of cosmogenic nuclides, primarily $^{10}$Be, in quartz-bearing river sand, has allowed for quantitative determination of landscape mass loss rates (hereafter, erosion rates) at a basin scale. Paired measurements of in situ cosmogenic $^{26}$Al and $^{10}$Be in sediment are less common but offer insight into the integrated exposure and burial history of sediment moving down slopes and through drainage basins. Prolonged burial ($>10^5$ years), a violation of assumptions underlying erosion rate calculations, is indicated by higher $^{26}$Al-based than $^{10}$Be-based erosion rates due to preferential loss of shorter-lived $^{26}$Al by decay when quartz is at least in part shielded from cosmic rays.

Here, we use a global compilation of $^{26}$Al and $^{10}$Be data generated from quartz-bearing fluvial sediment samples (n = 766, including 117 new measurements) to calculate the discordance between erosion rates derived from each nuclide. We find that over 30% of samples (n = 234) exhibit discordance ($> 2\sigma$ analytical uncertainty) between erosion rates derived from $^{10}$Be and $^{26}$Al, indicating sediment histories that include extended burial during residence on hillslopes and/or in the fluvial system after or during initial near-surface exposure. Physical basin parameters such as basin area, slope, and tectonic activity exhibit significant correlation with erosion rate discordance whereas climatic parameters have weak correlation, allowing us to infer the likelihood of sediment burial during transport in different geomorphic settings.

Paired $^{26}$Al and $^{10}$Be analyses in detrital fluvial samples provide a window into watershed processes, elucidating landscape behaviour at different spatial scales and allowing a deeper understanding of both sediment routing systems and whether methodological assumptions are violated. Although previous studies have found $^{26}$Al/$^{10}$Be erosion rate discordance to be common in the world's largest drainage basins, our analysis suggests that such discordance also occurs regularly in basins as small as 1,000 km$^2$, indicating that sediment storage mechanisms are more complex than suggested by simple floodplain-area scaling laws. Moderately sized basins (1,000 – 10,000 km$^2$) with low average slopes in tectonically quiescent terrains appear conducive to extended sediment storage; thus, erosion rates from such basins are lower limits due to nuclide decay during storage. We find that sediment sourced from smaller, steeper basins in tectonically active regions is more likely to have similar $^{10}$Be and $^{26}$Al erosion rates indicative of limited storage and is thus more likely to provide reliable erosion rates.

## 1 Introduction

Fluvial sediments are a rich source of information about the upstream sediment routing system, which encompasses sediment generation, transport, and storage processes (Romans et al., 2016; Tofelde et al., 2021). For example, in situ cosmogenic $^{10}$Be measurements of quartz isolated from fluvial sediments are used to estimate basin-averaged erosion rates. The application of this method in thousands of drainage basins around the world has provided valuable insights into physical and climatic controls on erosion (von Blanckenburg, 2005; Codilean et al., 2022; Portenga and Bierman, 2011; Schaefer et al., 2022). Such analyses assume an upstream sediment history in which material was generated through steady exhumation on hillslopes and then transported rapidly through fluvial networks, experiencing negligible storage while in transit (Bierman and Steig, 1996; von Blanckenburg, 2005; Granger et al., 1996; Granger and Schaller, 2014; Schaefer et al., 2022). Although erosion rates are now commonly measured, few studies have assessed the underlying assumptions of the technique and how often those assumptions are violated.

Sediment grains in fluvial systems can have a wide range of idiosyncratic transport and storage histories potentially spanning more than $10^6$ years in large basins, as shown by cosmogenic nuclide analyses in modern fluvial sediments (Fülöp et al., 2020; Repasch et al., 2020; Wittmann et al., 2011), volumetric and geochemical analyses of valley fills (Blöthe and Korup, 2013; Jonell et al., 2018; Munack et al., 2016), and sediment transport models (Carretier et al., 2020). These complex sediment histories, along with the protracted sediment lag times, may confound reliable interpretation of upstream processes (Allen, 2008; Jerolmack and Paola, 2010). Sediment samples used for analysis of cosmogenic nuclides are typically amalgamations of thousands of grains, each of which has its own unique history.

Measuring multiple in situ cosmogenic radionuclides with different half-lives is a promising approach for discerning fluvial sediment histories (Codilean and Sadler, 2021; Schaefer et al., 2022). Calculating ratios between multiple cosmogenic radionuclides has provided insight into sediment provenance (e.g., Cazes et al., 2020) and storage histories (e.g., Wittmann et al., 2011; Fülöp et al., 2020; Ben-Israel et al., 2022) in river systems around the world. Such studies have helped test hypotheses about sediment dynamics in river basins, including that the integrated storage duration experienced by sediments on hillslopes and in floodplains is generally greater in larger basins (Wittmann et al., 2020), in post-orogenic regions (Cazes et al., 2020; Struck et al., 2018), and in arid regions (Makhubela et al., 2019). However, such hypotheses have yet to be tested on a global scale and questions remain, such as whether sediment storage duration scales with physical and/or climatological basin metrics.

In this study, we compiled measurements of paired in situ $^{26}$Al and $^{10}$Be concentrations in detrital fluvial sediment from around the world (n = 766, including 117 new $^{26}$Al measurements on archived samples with previously published $^{10}$Be measurements) to test for the existence and likelihood of fluvial sediment storage across a wide range of physical and climatological drainage basin settings. We account for localized differences in nuclide production ratios to facilitate comparison across the world and use a variety of statistical tests to assess relationships between isotope concentrations and basin-scale landscape and climate parameters. Such a global description provides insight into the complexity of river sediment transport and storage and allows us to evaluate the validity of

assumptions inherent to the widely-used, basin-scale cosmogenic nuclide erosion rate method (von Blanckenburg,
2005; Granger and Schaller, 2014; Schaefer et al., 2022).

**2 Background**

**2.1 Sediment system dynamics and landscape change**

Fluvial sediments are products of hillslope processes and are moved through sediment routing systems. These

systems generally encompass regions of net sediment generation through bedrock weathering, regolith production,
and sediment export from hillslope source zones (Allen, 2017). This detrital material is then transported by fluvial
systems through riverine transfer zones and deposited in detrital sink zones (Schumm, 1977). Depending on the
geometry of the riverine transfer zone, sediment storage may be transient (e.g., steep bedrock streams) or long
lasting (e.g., lowland alluvial rivers). The extent and duration of storage in floodplains and sedimentary basins is an
important control on weathering (e.g., Campbell et al., 2022; Dosseto et al., 2014) as well as on both the production
of cosmogenic nuclides in sediments near the surface and the decay of those radionuclides if sediment is buried (Lal,

1991).

Understanding rates, controls, and dynamics of sediment generation and transport is important for quantifying

landscape change over time and space (Allen, 2008; Romans et al., 2016). In many routing systems, river
morphology (Langbein and Leopold, 1964; Leopold and Wolman, 1960) and floodplain volume (e.g., Otto et al.,
2009) are determined by the sediment mass flux out of source zones, the rate of transit through transfer zones, and
the accommodation space available for sediment storage. Changes to rates of sediment generation or transfer,
primarily driven by tectonic or climatic forcings (Romans et al., 2016), can thus affect the behaviour of both
sediment-supplying hillslopes and riverine transfer zones. Identifying such changes over space and through time is
an important objective of geomorphological research and has prompted the development of tracer and rate-
determining detrital geochronologic methods including measurements of cosmogenic nuclides, fission tracks, fallout
radionuclides, and U/Th/He in various mineral phases (Allen, 2017).
**2.2 Interpreting landscape processes from cosmogenic nuclides**

The application of cosmogenic nuclide analyses to fluvial sediments, first using single nuclides (Bierman and

Steig, 1996; Brown et al., 1995; Granger et al., 1996) and later paired nuclides (e.g., Clapp et al., 2000, 2001), has
significantly advanced our understanding of geomorphology and sediment routing systems at a variety of spatial and
temporal scales (e.g., Bierman and Nichols, 2004; von Blanckenburg, 2005; Codilean et al., 2021; Portenga and
Bierman, 2011; Willenbring et al., 2013; Wittmann et al., 2020). Key to the interpretation of measured nuclide
concentrations is a quantitative understanding of nuclide production and decay rates throughout the basin from
which the sediment is derived.

The ratio of $^{26}$Al to $^{10}$Be at production is ~6.8 at low and mid latitudes (Balco et al., 2008), but there are subtle

influences of latitude and altitude on that ratio (Argento et al., 2015; Halsted et al., 2021; Lifton et al., 2014).
Nuclide production decreases exponentially with depth below Earth's surface such that once sediment is buried
more than a meter or two, decay, rather than production systematics, controls the evolution of the $^{26}$Al/$^{10}$Be ratio
over time (Granger, 2006; Wittmann and von Blanckenburg, 2009).
Landscapes lose mass by both chemical and physical processes. The sum of these processes is referred to as
denudation and includes total mass loss integrated over depth. Mass loss rates inferred from cosmogenic nuclide
concentrations in sediment have most often referred to as erosion rates (Bierman and Steig, 1996; Granger et al.,
1996; Lal, 1991; VanLandingham et al., 2022) and we adopt that convention in this paper. We do this because our
data set includes numerous samples from parts of the world where there is deep chemical weathering (the tropics
and unglaciated, low-slope temperate regions). In these areas, mass loss through dissolution and groundwater export
extends many meters below the penetration depth of the cosmic ray neutrons responsible for most $^{10}$Be and $^{26}$Al
production. Such export of mass in solution is not reflected in the concentration of in situ produced cosmogenic
nuclides, which are only sensitive to mass loss in the uppermost few meters of Earth's dynamic surface (e.g.,
Campbell et al., 2022).
**2.2.1 Basin-scale erosion rates from single-nuclide measurements**
Basin-scale erosion rates have been estimated around the world by measuring the concentration of a single
cosmogenic nuclide, most often in situ $^{10}$Be, in samples of amalgamated river sediment (Bierman and Steig, 1996;
Brown et al., 1995; Codilean et al., 2022; Granger et al., 1996; Portenga and Bierman, 2011). Sediment grains
accumulate $^{10}$Be during exhumation and at the surface in source zones, with the nuclide concentration within grains
being proportional to the residence time of grains on hillslopes (Heimsath et al., 1997; Jungers et al., 2009). When
collecting a sample of fluvial sediment downstream, it is assumed that such a sample represents the average nuclide
concentration in grains sourced from all sediment-generating hillslopes within a basin (Bierman and Steig, 1996;
Granger et al., 1996; Brown et al., 1995).
Accuracy of basin-scale erosion rate calculations depends upon the validity of several assumptions about
sediment generation and transport that cannot be tested with single-nuclide analyses: that sampled grains were
steadily exhumed on hillslopes in sediment source zones, are well mixed, and are transported rapidly through fluvial
networks such that nuclide production and decay in the transport zone is minimal (Bierman and Steig, 1996;
Granger et al., 1996; Brown et al., 1995). This last assumption is most likely to be valid if the volume of sediment
stored in the system is small in comparison to the volume of sediment generated and transported through the system
on timescales relevant to $^{10}$Be production and decay (millennia; Granger et al., 1996).
**2.2.2 Sediment routing dynamics from paired $^{10}$Be and $^{26}$Al**
In situ $^{10}$Be and $^{26}$Al are the most commonly analysed cosmogenic nuclide pair in river sediment, with
measurements having started in the late 1990s (Bierman and Caffee, 2001; Clapp et al., 2000, 2001, 2002; Heimsath
et al., 1997; Nichols et al., 2002). Their popularity reflects the relative ease of extracting this isotope pair from the
same aliquot of quartz, the wide distribution of quartz across landscapes, and because of their contrasting half-lives
(1.4 My and 0.7 My, respectively, Chmeleff et al., 2010; Korschinek et al., 2010; Nishiizumi, 2004). When sediment
is buried, the shorter-lived $^{26}$Al is preferentially lost as decay exceeds production, and the $^{26}$Al/$^{10}$Be ratio in quartz
lowers over time (Balco and Rovey, 2008; Granger, 2006).

$^{26}$Al/$^{10}$Be ratios lower than those at production have been used as isotopic indicators of sediment storage and

subsequent remobilization in catchments across the world, ranging from arid (Bierman et al., 2001; Bierman and
Caffee, 2001; Clapp et al., 2002; Kober et al., 2009) to tropical (Campbell et al., 2022; Wittmann et al., 2011)
climates and in small (Clapp et al., 2000, 2001) to very large (Ben-Israel et al., 2022; Fülöp et al., 2020; Hidy et al.,
2014; Wittmann et al., 2020; Wittmann and von Blanckenburg, 2016) basins. However, in some studies, lowered
$^{26}$Al/$^{10}$Be ratios were attributed to laboratory errors (Insel et al., 2010; Walcek and Hoke, 2012; Hattanji et al., 2019)
or incorporation of meteoric $^{10}$Be (Corbett et al., 2022; Moon et al., 2018) and disregarded.

In this study, sediment burial (and resulting preferential loss of shorter-live $^{26}$Al by decay) is reflected by the

discordance between erosion rates calculated from $^{10}$Be ($E_{Be}$) and $^{26}$Al ($E_{Al}$), the calculation of which normalizes
spatial variations in the $^{26}$Al/$^{10}$Be surface production ratio and accounts for differential nuclide decay during
prolonged surface exposure in very slowly eroding terrains. Thus, calculating erosion rate discordance rather than
using nuclide concentration ratios facilitates comparisons between basins across the world and is sensitive only to
nuclide decay caused by sediment burial after initial exposure, rather than decay that occurs during prolonged
surface or near-surface exposure.

If sediment is transferred from slopes into channels and transported through the channel network without

extended burial, then erosion rates calculated from the concentration of each nuclide should be coincident ($E_{Be}$ =
$E_{Al}$). Discordance between erosion rates calculated from the two nuclides (unless it is caused by laboratory errors)
reflects preferential loss of $^{26}$Al when and where decay exceeds production, in which case $E_{Be} < E_{Al}$. This occurs
when sediment is stored below the surface (>2m) and for extended periods (>$10^5$ years) after initial surface exposure
on hillslopes or in floodplains (Fig. 1).

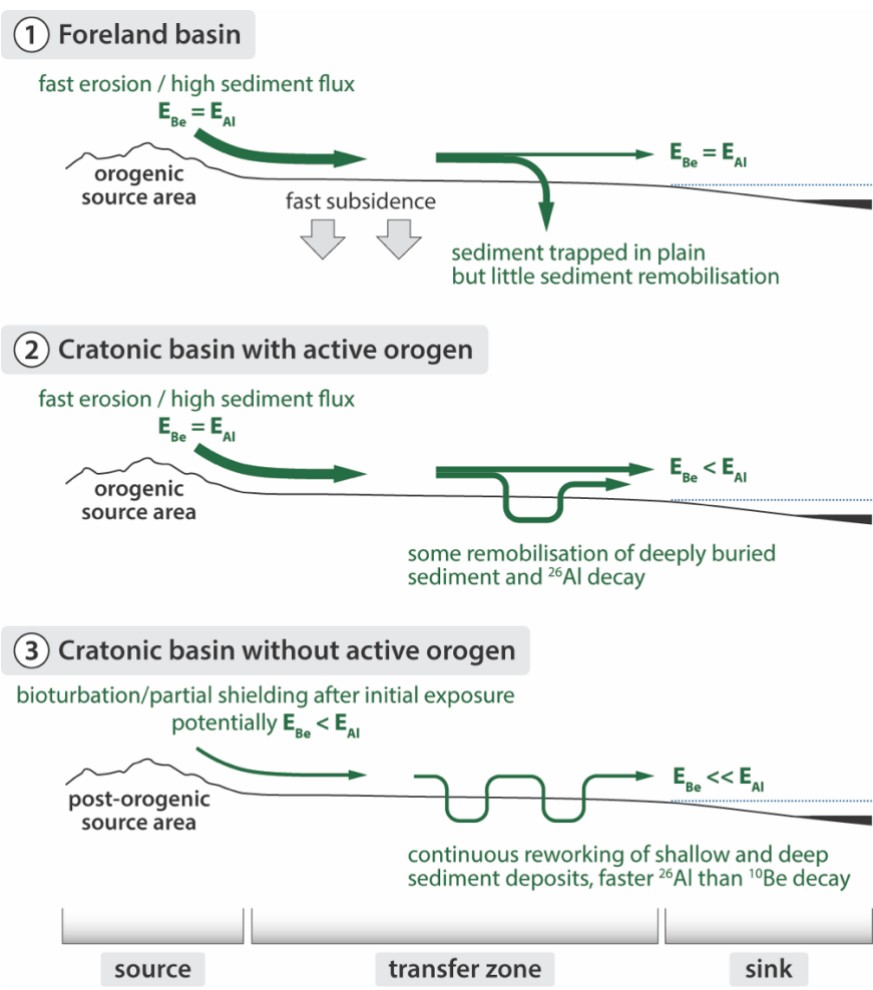


**Figure 1: Effects of storage in sediment source and/or transfer zones on [10]Be and [26]Al-based erosion rates measured in detrital quartz grains. In Panel 1, rapid erosion rates in the source zone and limited remobilization of stored sediment in the transfer zone result in detrital sediment with concurrent erosion rates ($E_{Be}/E_{Al} = 1$). In Panel 2, rapid erosion rates in the source zone and some remobilization of stored sediment in the transfer zone result in detrital sediment with erosion rate discordance ($E_{Be}/E_{Al} < 1$), although prolonged sediment storage ($>10^5$ years) is necessary for erosion rate discordance to be measurable. In Panel 3, slow erosion rates in the source zone and remobilization of stored sediment in the transfer zone result in detrital sediment with substantial erosion rate discordance ($E_{Be}/E_{Al} << 1$). This figure is based on Figure 6 in Wittmann et al. (2016).**

Floodplain sediment storage of $<10^5$ years has minimal effect on $E_{Be}/E_{Al}$ in sediment grains (Wittmann and von Blanckenburg, 2009), but during prolonged ($>10^5$ years) storage, especially at depths below which most nuclide production by spallation occurs (> several hundred g cm$^{-2}$), the $E_{Be}/E_{Al}$ of amalgamated samples can lower sufficiently that the lowering can be detected with confidence in quartz containing moderate to high concentrations of these nuclides (Fig 1). In slowly-eroding terrains (<10 m My$^{-1}$), long subsurface sediment residence times on hillslopes after initial exposure, due to vertical mixing, can lead to erosion rate discordance in sediment source areas due to preferential [26]Al decay before regolith reaches the channel (Fig. 1; Makhubela et al., 2019; Struck et al., 2018). The rate of $E_{Be}/E_{Al}$ lowering is depth-dependent, the ratio decreases more rapidly with increased sediment burial depth as nuclide production rates decrease.

Re-introduction of stored sediment with low $E_{Be}/E_{Al}$ back into the active channel will lower the average $E_{Be}/E_{Al}$

of fluvial sediment in transport (Wittmann et al., 2009; Fig. 1). Geomorphic processes responsible for sediment
reworking in transfer zones vary widely depending on basin morphology, tectonics, and climatology. Extensive
sediment storage followed by remobilization is documented in meandering, low-lying, tropical river systems
(Wittmann et al., 2011), arid river systems that source sediment from sand dunes containing long-buried sediments
(Eccleshall, 2019; Vermeesch et al., 2010), hydrologically-variable basins where flood events remobilize vertically-
accreted floodplain deposits (Codilean et al., 2021), and formerly glaciated basins where sediments were repeatedly
covered by ice (Jautzy et al., 2024). While old, deeply-buried deposits typically have low nuclide concentrations and
thus less influence on the average $E_{Be}/E_{Al}$ when mixed with active channel sediment in small amounts, high flow
events may re-mobilize substantial volumes of long-buried sediment and have a significant impact on nuclide
concentrations (e.g., Codilean et al. 2021; Wittmann et al., 2011) and calculated $E_{Be}/E_{Al}$.
**3 Methods**
**3.1 Study Design – Approach and Limitations**

In this study, we use a compilation of previously-published (n = 649) and new (n = 117) paired [10]Be and [26]Al

concentration measurements in fluvial sediments to assess storage and remobilization during sediment generation
and/or transport. We calculate nuclide-specific erosion rates and use the agreement or discordance between these
rates to identify burial during transport. We measure the morphometric and climatological properties of basins from
which the sampled sediments derive and use a variety of statistical analyses to assess if basin properties are
correlated with cosmogenic indications of burial. Then, we consider geomorphic mechanisms to explain observed
correlations and discuss the implications of our results for the widely-used basin-averaged [10]Be erosion rate method.

Measured [26]Al and [10]Be alone cannot quantify sediment storage durations or identify specific geomorphic

histories for each sample because sediment samples are mixtures of grains with different histories and the inverse
solutions are non-unique (Bierman and Steig, 1996; von Blanckenburg, 2005; Brown et al., 1995; Granger et al.,
1996; Schaefer et al., 2022). The rate of $E_{Be}/E_{Al}$ lowering in stored sediment is depth-dependent (Wittmann and von
Blanckenburg, 2009); thus, the mixing of grains with different storage depth and time histories, and consequently
varying histories and duration of nuclide decay and production, precludes accurate estimations of storage duration.
Although we identify basin properties that correlate with isotopic indications of burial and storage, the identification
of specific processes responsible for storage and subsequent remobilization will differ on a case-by-case basis.
**3.2 Data sources**

We used two data sources: measurements in reported published studies (n = 649) and [26]Al and [10]Be

concentrations from new [26]Al measurements made on samples archived at the University of Vermont (UVM) that
had previously published [10]Be concentrations (n = 117). For all samples, we normalized originally-reported [10]Be
concentrations to the 07KNSTD standard (Nishiizumi et al., 2007) and [26]Al concentrations to the KNSTD standard
(Nishiizumi, 2004) using conversion factors based on the original AMS standards used for normalization (Table S1;
Balco et al., 2008; Nishiizumi et al., 2007).

### 3.2.1 Sources of previously published paired $^{26}$Al and $^{10}$Be measurements

We sourced data from the OCTOPUS database (Codilean et al., 2018, 2022; Codilean and Munack, 2024) for
previously-published paired $^{26}$Al and $^{10}$Be measurements from fluvial sediments around the world with robust
documentation of processing methods, including the Al and Be standards used during AMS measurements (n =
555). We also compiled samples from studies that had not yet been added to the OCTOPUS database at the time of
writing (n = 94; Wang et al., 2017; Adams and Ehlers, 2018; Mason and Romans, 2018; Moon et al., 2018; Hattanji
et al., 2019; Hubert-Ferrari et al., 2021; Yang et al., 2021; Zhang et al., 2021; Ben-Israel et al., 2022; Zhang et al.,
2022; Jautzy et al., 2024). Previously-published samples were processed at numerous laboratories, including at
UVM, and were analyzed at several AMS facilities (sources, raw data, and AMS facilities for previously published
samples are reported in Table S1).

### 3.2.2 Sample processing for new $^{26}$Al measurements

Samples with new $^{26}$Al measurements come from a wide range of locations but were processed entirely at
UVM between 2009 and 2019. These archived samples had previously undergone Be and Al extraction following
established methods (Corbett et al., 2016), but only had $^{10}$Be concentrations measured ($^{10}$Be concentration
measurements were originally reported in their source publications and are provided in Table S2). The Al-bearing
fraction of these archived samples, Al and Be having been separated by column chromatography during the original
sample processing for $^{10}$Be analysis (Corbett et al., 2016), were stored as Al hydroxide gels.
We re-dissolved the gels into a chloride liquid form using 1 mL of 6 mol/L hydrochloric acid and allowed the
gels to sit in acid for several weeks. When completely dissolved, we added 4 mL of water to each sample to create a
1.2 mol/L hydrochloric acid solution for column chromatography and centrifuged the samples to remove any
lingering undissolved material. We removed $^{26}$Mg, an isobar of $^{26}$Al, via column chromatography and then followed
the methods outlined in Corbett et al. (2016) to convert samples into an Al oxide powder mixed with Nb for
$^{26}$Al/$^{27}$Al measurement via accelerator mass spectrometry (AMS).
$^{26}$Al/$^{27}$Al ratios for these re-processed samples were measured using AMS between 2019 and 2021 at the
Purdue Rare Isotopes Measurement Laboratory (PRIME), where the addition of a gas-filled magnet to the AMS has
significantly reduced $^{26}$Al measurement uncertainties (Caffee et al., 2015). Samples were measured against primary
standard KNSTD with a $^{26}$Al/$^{27}$Al ratio of 1.818 x 10$^{-12}$ (Nishiizumi, 2004). We re-processed blanks that were
archived with the Al hydroxide gels from their original processing batches (n = 37) and blank-corrected samples by
subtracting the estimated $^{26}$Al atoms in the batch-specific blank from the total $^{26}$Al in the sample (Table S2). Where
the original batch blank was missing, likely due to others re-sampling Al gels from the batch prior to 2019, the
average $^{26}$Al/$^{27}$Al ratio from all re-processed blanks (2.37 +/- 1.84 x 10$^{-15}$; 1SD) was used to estimate a blank
correction. We propagated AMS $^{26}$Al/$^{27}$Al and blank measurement uncertainties in quadrature to quantify total $^{26}$Al
concentration uncertainty. All new [26]Al concentration, blank, and uncertainty measurements and calculations can be
found in Table S2.

**3.3 Calculating [10]Be and [26]Al-derived erosion rates and erosion rate discordance.**

We use the erosion rate calculator formerly known as CRONUS v3 (Balco et al., 2008) with the nuclide-
specific LSDn scaling scheme (Lifton et al., 2014) to calculate $E_B$ and $E_{Al}$. The LSDn scaling scheme depicts spatial
variations in the [26]Al/[10]Be surface production ratio (Halsted et al., 2021) and thus calculated $E_B$ and $E_{Al}$ values are
normalized to local nuclide-specific production rates to facilitate comparisons across the world. We assumed no
shielding and estimate spatially-averaged basin altitude scaling factors using an iterative process that identifies the
atmospheric pressure value best matching the spatially averaged Lal/Stone production rate in each basin, a more
computationally efficient method than pixel-based approaches for this large compilation and with nearly
indistinguishable results (Codilean and Munack, 2024). We propagated 'internal' uncertainties (i.e., analytical
uncertainties) of $E_{Be}$ and $E_{Al}$ estimates in quadrature to quantify the 1-sigma (1σ) uncertainty of $E_{Be}/E_{Al}$.
A $E_{Be}/E_{Al}$ value indistinguishable from 1 (considering 2σ uncertainties) is consistent with a history without
burial (but does not necessarily preclude burial and then re-exposure). A $E_{Be}/E_{Al}$ distinguishably lower than 1 is
consistent with a history including burial and remobilization of sediment back into the active channel. $E_{Be}/E_{Al}$ values
distinguishably higher than 1 are theoretically impossible and likely indicate laboratory processing and/or
measurement errors.

**3.4 Quantifying basin parameters**

For each basin, we calculated [10]Be and [26]Al-derived erosion rates, mean basin slope, basin area, local relief
using a 2 km radius circular moving window, mean annual precipitation, aridity, tectonic activity, dominant
lithology, likelihood of stream flow intermittence, glacial cover at the Last Glacial Maximum, and present-day ice
cover (data sources and detailed methods are reported in the Supplementary Material). We created basins shapefiles
by delineating watersheds upstream of sediment sampling locations (following the procedures used in the
OCTOPUS database; Codilean et al., 2022) and used these shapefiles to calculate zonal statistics within each basin.
We determined all sampling locations from the source publications or through personal correspondence with the
papers' authors. We treated nested basins individually, such that a sample collected in an upstream tributary basin
has a separate basin shapefile from the larger, downstream sample with a basin encompassing all upstream
tributaries.

**3.5 Statistical analyses**

We used hypothesis testing methods to determine if physical or climatological characteristics of sample basins
correlate significantly with calculated $E_{Be}/E_{Al}$ values. We used correlation analyses between $E_{Be}/E_{Al}$ values and
numerical basin parameters (latitude, mean erosion rate, area, mean area, mean slope, mean local relief, annual
precipitation, aridity index, intermittent flow probability, percent cover by both Last Glacial Maximum and present
ice, and hypsometric integral) and checked for cross-correlation between all basin parameters. We log-transformed
basin areas and basin-averaged $^{10}$Be erosion rates prior to correlation analyses to normalize their skewed distribution
(Fig. 4) and used the non-parametric Spearman's Correlation Coefficient to evaluate the strength of correlations due
to the lingering non-normality of some basin parameter distributions.
We used a forward stepwise regression analysis as in Portenga and Bierman (2011) to create a multi-variate
linear model relating $E_{Be}/E_{Al}$ values to basin parameters. This analysis considers all basin parameters but only fits a
regression through those that are most statistically important as defined by the change in p-value of the model F-
statistic when adding or removing each parameter. We set the probability to enter as $p < 0.05$ and the probability to
leave as $p > 0.1$.
We use one-way analysis of variance (ANOVA) and Tukey multiple comparison of means testing (Abdi and
Williams, 2010) to assess the magnitude and statistical significance of $E_{Be}/E_{Al}$ value differences between categorical
variables (tectonic activity, dominant lithology, region) and to identify threshold values for $E_{Be}/E_{Al}$ differences based
on basin areas and hypsometric integrals. We ran the same analyses using the Kruskal-Wallis H test for multiple
comparison of medians (MacFarland and Yates, 2016) and obtained nearly identical results to the Tukey MCM
testing; we report only the mean results. We used the python libraries *pandas*, *matplotlib*, *cartopy*, *numpy*, *seaborn*,
*scipy*, and *statsmodels* to perform all statistical analyses (except for the forward stepwise regression analysis) and
create figures, and a Jupyter notebook with coding for all analyses (including the median analyses) is included in the
Supplementary Material. We used MATLAB to perform the forward stepwise regression analysis using the
'*stepwiselm*' function; a copy of this script can be found in the Supplementary Material.
**4 Results**
**4.1 Dataset statistics**
The compilation of basins assembled here (n = 766) has near-global coverage, although there are fewer data
from low-latitude regions, especially at high elevations (Figs 2 and 3). Most basins are < 100,000 km$^2$ (n = 677),
while a small number (n = 25) are very large (>1,000,000 km$^2$; Fig 2). The basins in the compilation encompass a
wide range of morphologic and climatic regimes (Fig 4). The distributions of most basin parameters are right-
skewed, with most basins having low-to-moderate slope, relief, and precipitation. The basins are underlain by a
variety of dominant lithologies and are split almost evenly between those that are tectonically active (n = 411) and
those that are post-orogenic (n = 355).

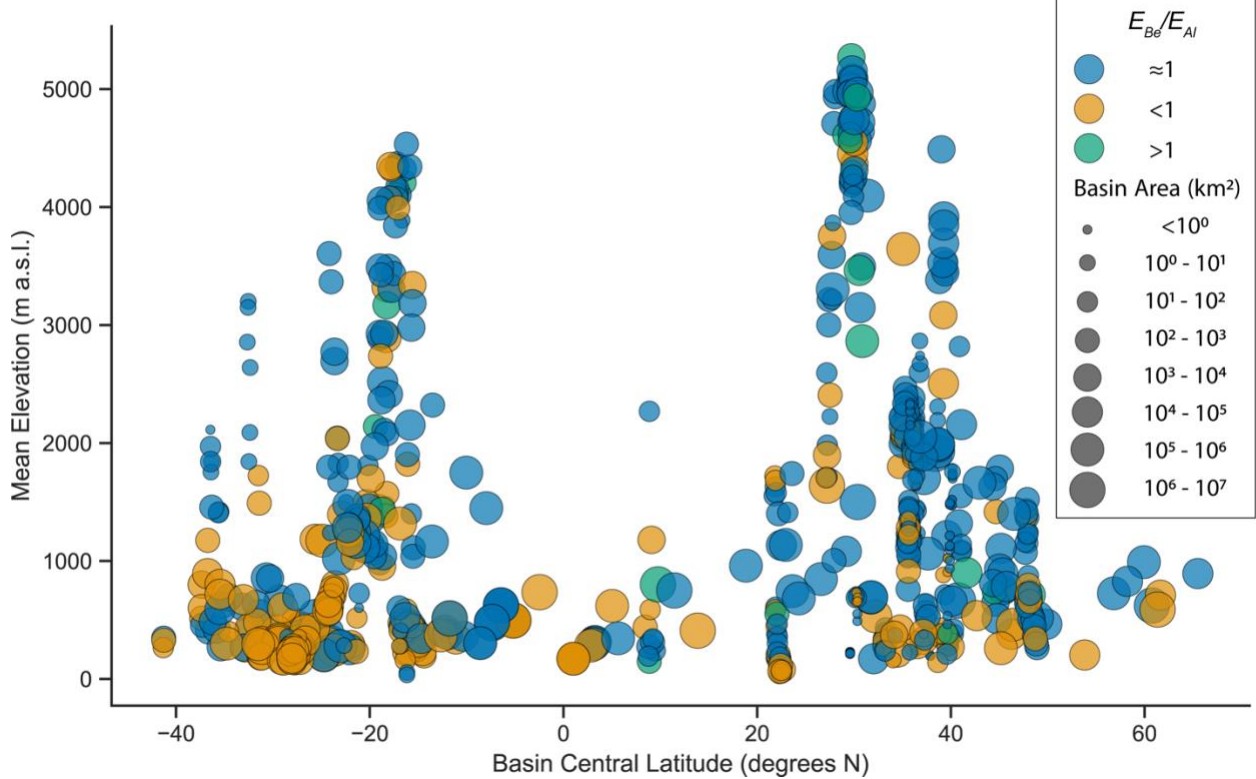

**Figure 2. Latitude and elevation distribution of basins in our compilation. Color coding indicates if calculated erosion rate**
**ratios are indistinguishable from 1 (considering 2σ analytical uncertainties), distinguishably lower than 1 or**
**distinguishably higher than 1. The marker size indicates source basin area.**

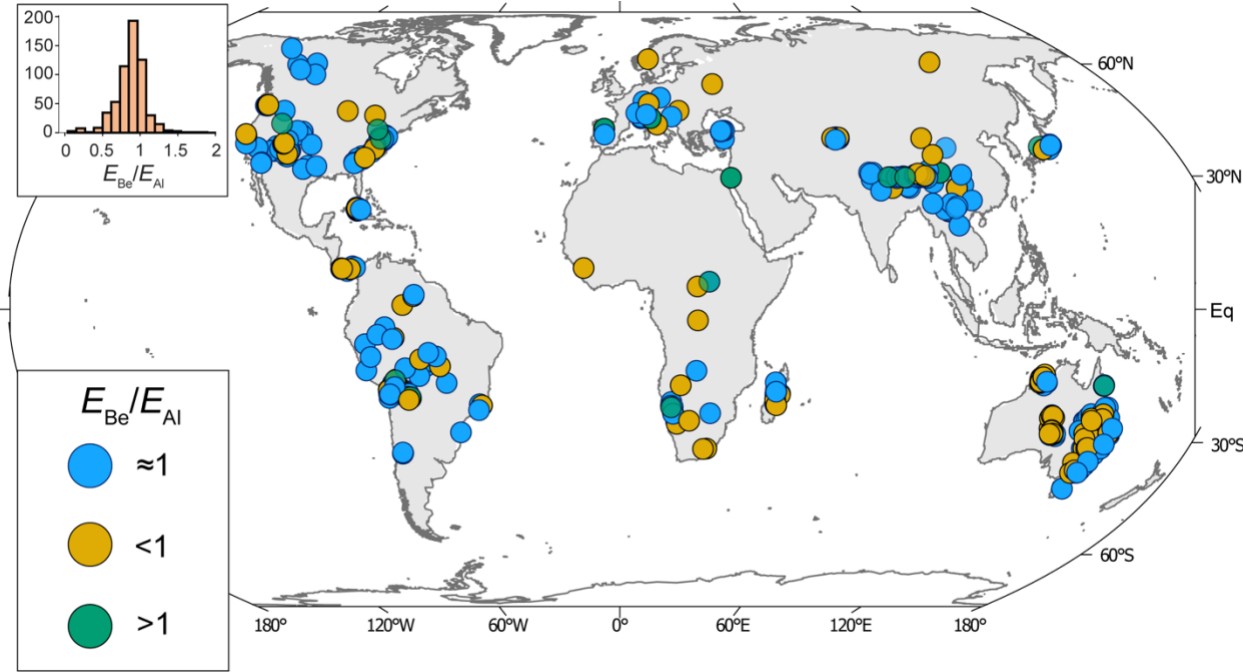

**Figure 3. Inset: Distribution of $E_{Be}/E_{Al}$ values. Main: Map of basin centroid locations color-coded by $E_{Be}/E_{Al}$ values.**

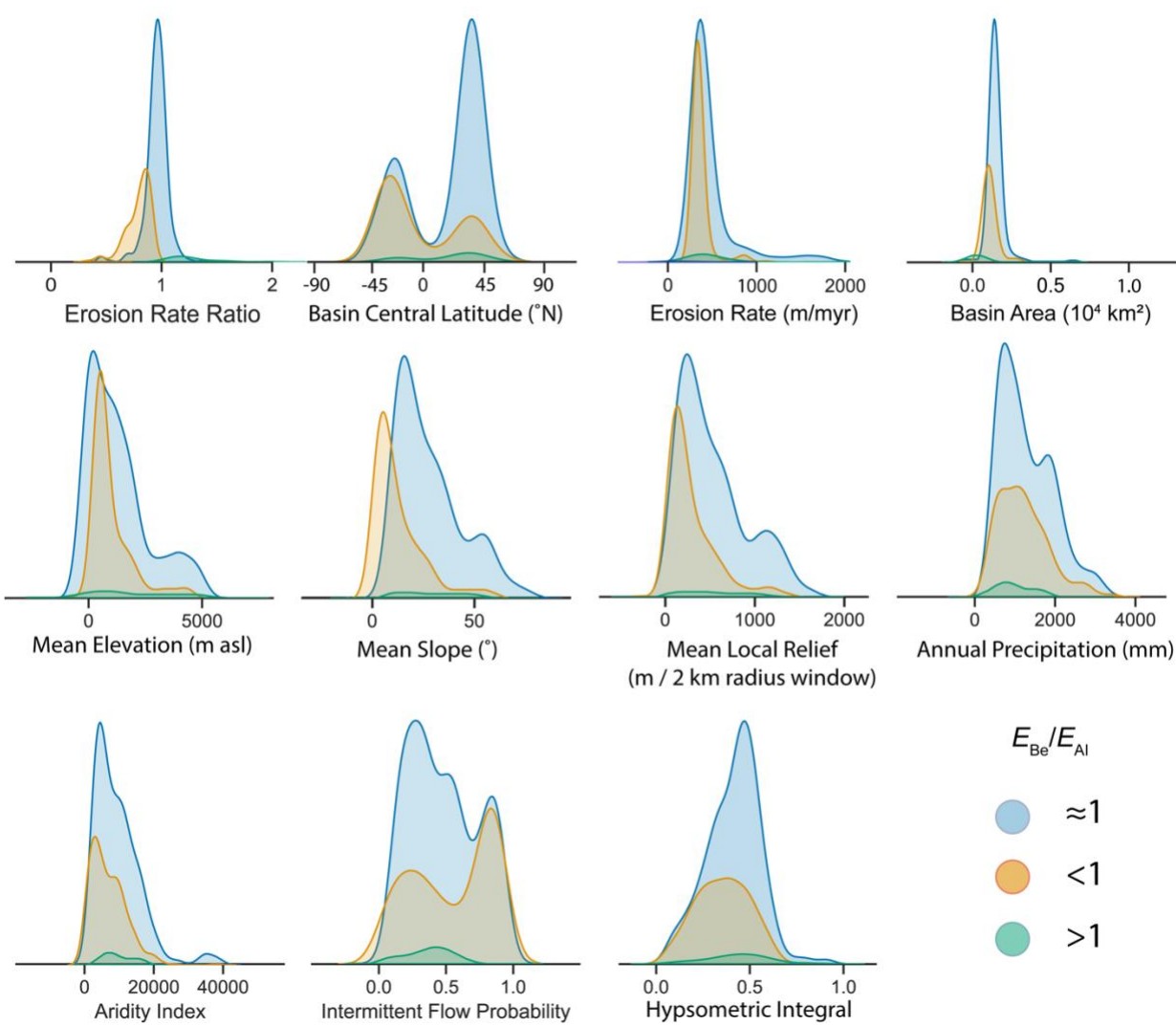


**Figure 4. Kernel density distributions of basin parameters with subdivision and color-coding based on $E_{Be}/E_{Al}$ values (see legend). Note that the erosion rate and basin area plots both feature extremely long tails on the high end and thus have x-axis limits imposed. Vertical axes on all plots are relative density values. Sources for all parameters and methods used in their calculations are provided in the Supplementary Materials.**

The population of $E_{Be}/E_{Al}$ values (n = 766) approximates a normal distribution with mean = 0.88 and SD = 0.21 (Fig 3 inset). Approximately 31% of the samples in the compilation (n = 238) have $E_{Be}/E_{Al}$ values that are distinguishably lower than 1 when considering 2σ analytical uncertainties, while ~3.5% of samples (n = 27) have $E_{Be}/E_{Al}$ values distinguishably higher than 1.

**4.2 Correlation analysis and stepwise regression**

Of the basin parameters, all but aridity index exhibit statistically-significant correlations with $E_{Be}/E_{Al}$ values (p < 0.05), although none of the correlations with $E_{Be}/E_{Al}$ are particularly strong ($r_s$ < 0.4; Figure 5).

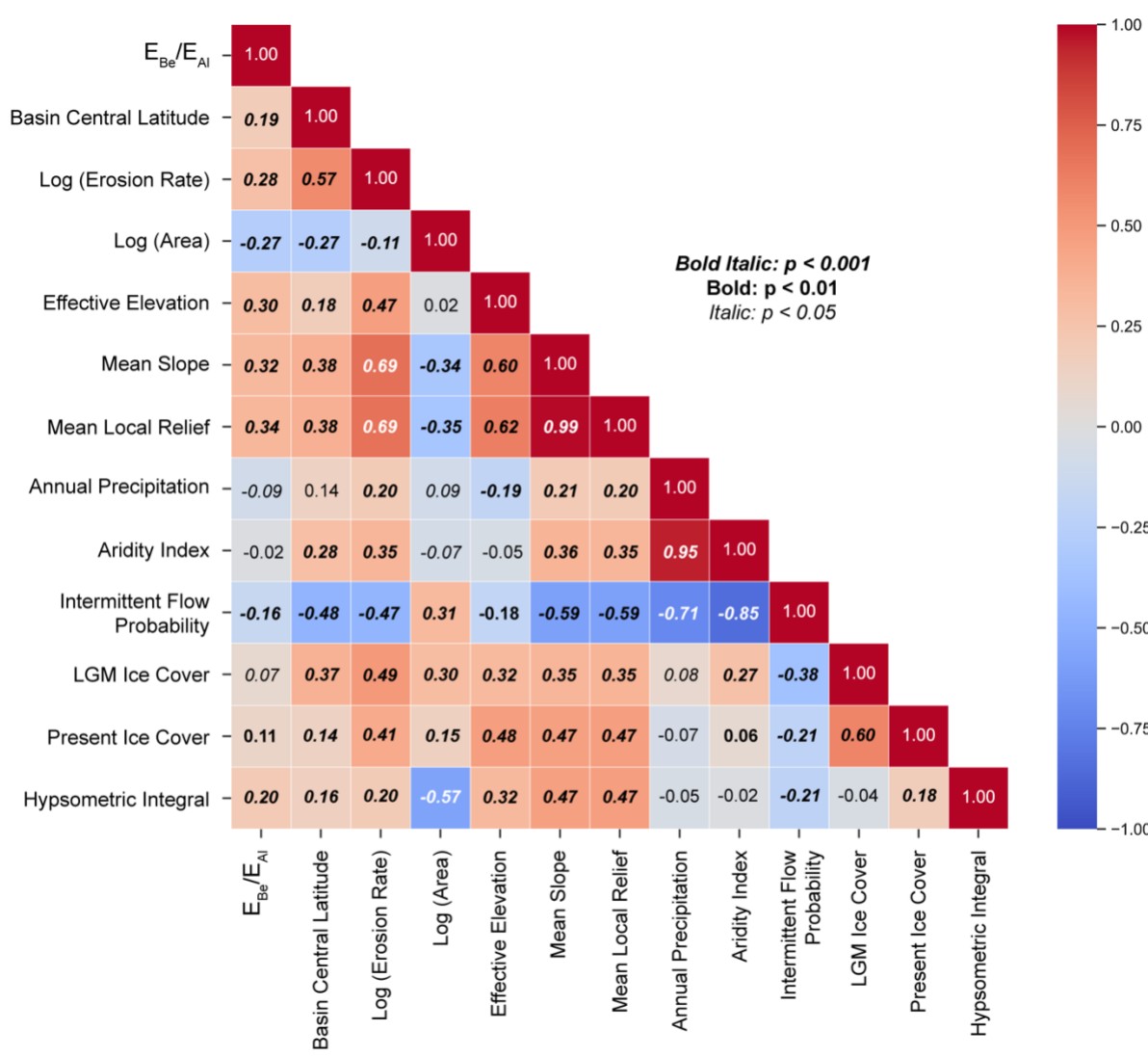

**Figure 5. Cross-correlation matrix for basin parameters and $E_{Be}/E_{Al}$ values. Color scale shows Spearman's Correlation Coefficient values and font styling indicates statistical significance (p value) of correlation coefficient.**

The best-fitting linear model from the forward stepwise regression analysis (Table 1) predicts a decrease in $E_{Be}/E_{Al}$ values with increasing basin area, decreasing basin-averaged erosion rate, decreasing basin mean elevation, and decreasing hypsometric integral. No other basin parameters improved this multivariate model and thus those parameters were removed during the stepwise regression analysis. This model represents a statistically-significant improvement over a constant model ($p \ll 0.001$), although a low reduced chi-squared statistic (0.048) suggests that it may overfit the data.

**Table 1: Summary of linear model ($E_{Be}/E_{Al} \sim \beta + X + Y$) output from forward stepwise regression analysis**

|  | **Estimate** | **SE** | **tStat** | **p-value** |
|---|---|---|---|---|
| *(Intercept)* | 0.923 | 0.040 | 22.868 | 1.766e-88 |
| *Log(Area)* | -0.014 | 0.002 | -5.941 | 4.308e-09 |
| *Log(Erosion Rate)* | 0.017 | 0.005 | 3.597 | 3.422e-04 |
| *Mean Elevation* | 3.215e-05 | 7.035e-06 | 4.571 | 5.678e-06 |
| *Hypsometric Integral* | 0.163 | 0.072 | 2.266 | 2.376e-02 |

*Number of observations: 765, Error degrees of freedom: 760*
*Root Mean Squared Error: 0.207, R-squared: 0.111, Adjusted R-Squared: 0.107*
*F-statistic vs. constant model: 23.8, p-value = 1.49e-18*
*Reduced Chi-Square: 0.048*
**4.3 ANOVA testing**
ANOVA testing offers more granular insight into the decline of $E_{Be}/E_{Al}$ values with increasing basin area
and decreasing hypsometric integral, and among categorical basin parameters suggests that tectonic activity, but not
dominant lithology, has a significant correlation with measured $E_{Be}/E_{Al}$ values (Figure 6). Post-hoc tests using group
mean and median values produced nearly identical results; mean tests are shown here while the results from median
post-hoc tests are included in the supplementary information.

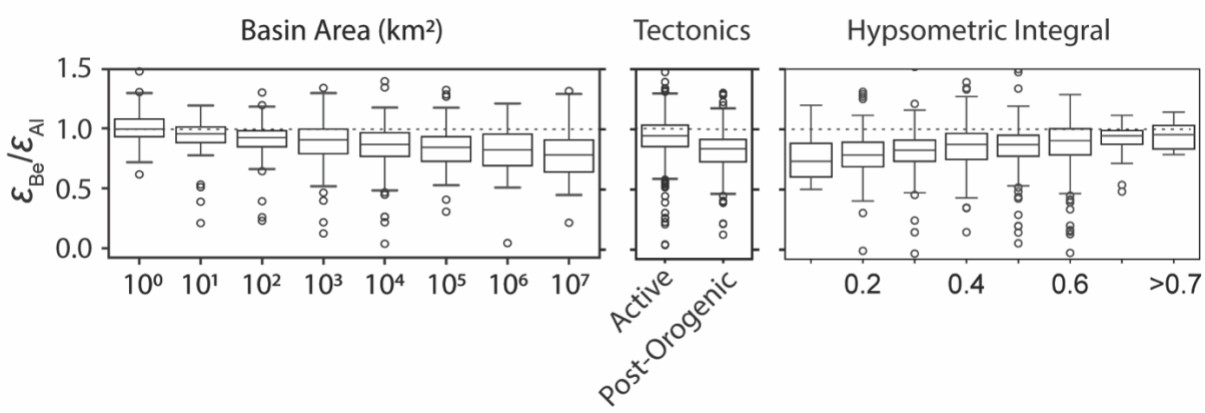


**Figure 6. One-way ANOVA results comparing $E_{Be}/E_{Al}$ values between basin area categories (left, basins in each category**
**have areas less than or equal to the label on the x-axis), basin tectonic activity (center), and dominant basin lithology**
**(right). In each plot, boxes show median (center line), 25th and 75th percentile values (box edges) and the maximum and**
**minimum non-outlier values (whiskers). $E_{Be}/E_{Al}$ values plotted as circles are considered outliers (more than 1.5x the**
**interquartile range). The dashed horizontal line in all plots is a reference line for $E_{Be}/E_{Al} = 1$. Note that n=8 samples have**
**$E_{Be}/E_{Al} > 1.5$ and are cropped out of this figure.**
With basin areas binned on a logarithmic base-10 scale, a decline in $E_{Be}/E_{Al}$ values with increasing basin
area is clear (Figure 6; Table 2). Very small basins ($\leq 1$ km$^2$) have a mean $E_{Be}/E_{Al}$ value of approximately 1 ($\mu = 0.96$
$\pm 0.20$, n = 83) while the largest basins ($>1,000,000$ km$^2$) have a mean $E_{Be}/E_{Al}$ value of $0.79 \pm 0.25$ (n = 25). We use
a multi comparison test to assess if $E_{Be}/E_{Al}$ mean values for each basin area category are significantly different than
the smallest basin group and find that basins larger than 1,000 km$^2$ have mean $E_{Be}/E_{Al}$ values less than 1. The
percentage of basins with $E_{Be}/E_{Al}$ values that are lower than 1 (considering $2\sigma$ uncertainties) increases from 13% in
the < 1 km$^2$ area bin to 40% in the $10^4$ km$^2$ area bin and remains above 35% for all larger basins (Table 2).
**Table 2: One-way ANOVA results comparing measured $E_{Be}/E_{Al}$ values between basin area categories. Note that the label**
**for each basin area category shows the upper limit for basin areas in that bin.**

| Basin Area (km$^2$) | n | $E_{Be}/E_{Al}$ Mean | $E_{Be}/E_{Al}$ S.D. | MCMean to $10^0$ km$^2$ basins, p-value* | % of Basins with $E_{Be}/E_{Al}$ < 1** |
|---|---|---|---|---|---|
| $10^0$ | 83 | 0.96 | 0.20 | - | 13 |
| $10^1$ | 68 | 0.93 | 0.28 | 0.39 | 15 |
| $10^2$ | 119 | 0.90 | 0.18 | 0.13 | 25 |
| $10^3$ | 180 | 0.88 | 0.22 | 0.02 | 32 |
| $10^4$ | 136 | 0.87 | 0.21 | <0.01 | 42 |
| $10^5$ | 91 | 0.83 | 0.22 | <0.01 | 42 |
| $10^6$ | 64 | 0.82 | 0.17 | <0.01 | 39 |
| $10^7$ | 25 | 0.79 | 0.25 | <0.01 | 36 |

*Shows p-value for Tukey multi-comparison of means test performed between basin area category and the smallest*
*basins (<$10^0$ km$^2$)*
***Including $2\sigma$ uncertainties*
Basin hypsometric integrals also have a statistically significant influence on $E_{Be}/E_{Al}$ based on ANOVA testing
(Figure 6; p = 0.007). Mature basins with low hypsometric integrals generally have lower mean $E_{Be}/E_{Al}$ values
compared to basins with high hypsometric integrals, but a multi comparison of means test demonstrates that these
differences are not statistically significant (Table 3). The percentage of basins with $E_{Be}/E_{Al}$ values that are lower than
1 (considering $2\sigma$ uncertainties) increases from 9% in the >0.7 hypsometric integral bin to >30% for basins with
hypsometric integrals <0.4 (Table 3).
**Table 3: One-way ANOVA results comparing measured $E_{Be}/E_{Al}$ values between hypsometric integral categories. Note that**
**the label for each hypsometric integral category shows the upper limit for the integral in that bin.**

| Hypsometric Integral | n | $E_{Be}/E_{Al}$ Mean | $E_{Be}/E_{Al}$ S.D. | MCMean to >0.7 basins, p-value* | % of Basins with $E_{Be}/E_{Al}$ < 1** |
|---|---|---|---|---|---|
| 0.1 | 22 | 0.81 | 0.21 | 0.09 | 36 |
| 0.2 | 60 | 0.83 | 0.24 | 0.07 | 48 |
| 0.3 | 115 | 0.84 | 0.20 | 0.08 | 43 |
| 0.4 | 164 | 0.89 | 0.23 | 0.38 | 34 |

| | | | | | |
|---|---|---|---|---|---|
| *0.5* | 215 | 0.89 | 0.22 | 0.39 | 26 |
| *0.6* | 147 | 0.89 | 0.22 | 0.35 | 22 |
| *0.7* | 31 | 0.95 | 0.13 | 0.93 | 23 |
| *>0.7* | 11 | 1.04 | 0.29 | - | 9 |

*\*Shows p-value for Tukey multi-comparison of means test performed between hypsometric integral category and the*
*category >0.7.*
*\*\*Including 2σ uncertainties*

We find that basins in tectonically active settings have higher $E_{Be}/E_{Al}$ values ($\mu = 0.93 \pm 0.22$, n = 411) than

post-orogenic basins ($\mu = 0.83 \pm 0.20$, n = 355); this difference is statistically significant ($p \ll 0.01$). Dominant
basin lithology has less influence on $E_{Be}/E_{Al}$ values (Table 4). Most lithologies have mean $E_{Be}/E_{Al}$ values that are
statistically indistinguishable from each other. The exception is basins composed primarily of unconsolidated
sediments, which have, on average, lower $E_{Be}/E_{Al}$ values than other lithologies ($\mu = 0.79 \pm 0.22$, n = 105). The
presence of glacial deposits in basins, here categorized as basins with more than 10% coverage by Last Glacial
Maximum ice (Ehlers et al., 2011), appears to have little influence on erosion rate discordance when considered at
this global scale; basins containing glacial deposits have an average $E_{Be}/E_{Al}$ value ($\mu = 0.90 \pm 0.25$, n = 117)
indistinguishable from those without glacial deposits ($\mu = 0.88 \pm 0.21$, n = 649).

**Table 4: Mean $E_{Be}/D\backslash E_{Al}$ values and standard deviations for dominant basin lithologies as defined in the GLiM database (Hartmann and Moosdorf, 2012)**

| Lithology | n | $E_{Be}/E_{Al}$ Mean | $E_{Be}/E_{Al}$ S.D. |
|---|---|---|---|
| *Acid Plutonic* | 134 | 0.93 | 0.24 |
| *Acid Volcanic* | 29 | 0.94 | 0.16 |
| *Basic Volcanic* | 14 | 0.92 | 0.20 |
| *Carbonate Sedimentary* | 28 | 0.94 | 0.39 |
| *Intermediate Volcanic* | 9 | 0.77 | 0.32 |
| *Metamorphic* | 104 | 0.90 | 0.17 |
| *Mixed Sedimentary* | 107 | 0.93 | 0.21 |
| *Pyroclastic* | 6 | 0.86 | 0.18 |
| *Siliciclastic Sedimentary* | 226 | 0.86 | 0.18 |
| *Unconsolidated Sediments* | 105 | 0.79 | 0.22 |

## 5 Discussion and implications

### 5.1 Prevalence and potential mechanisms causing complex sediment histories

We find widespread evidence of sediment histories that likely include extended sediment storage on timescales of $10^5 - 10^6$ years as indicated by $E_{Be}/E_{Al}$ values distinguishably lower than 1 (considering $2\sigma$ analytical uncertainties) in over 30% of sampled basins around the world (n = 238). This substantial number of low $E_{Be}/E_{Al}$ values rejects the null hypothesis ($p << 0.001$) of minimal nuclide decay due to sediment storage that is assumed in many single-nuclide erosion rate studies.

The occurrence and magnitude of depressed $E_{Be}/E_{Al}$ values is correlated with several basin morphological parameters, suggesting a systematic and thus predictable relationship between basin morphology and sediment history. Although most physical basin parameters exhibited statistically significant correlations with measured $E_{Be}/E_{Al}$ values (Fig. 2), widespread cross-correlations exist between these parameters and suggest several basin characteristics considered together are more likely to predict sediment histories including extended burial. The number of $E_{Be}/E_{Al}$ values distinguishably higher than 1 (n = 27) is within the range expected due to Poisson-distributed measurement uncertainties and is not statistically significant ($p >> 0.05$).

Stepwise linear regression and ANOVA testing suggests that basin area has the single largest influence on $E_{Be}/E_{Al}$ values (Figs 5 and 6, Tables 1 and 2). The scaling of sediment storage duration with basin area is expected (Pizzuto, 2020), with extended burial leading to significant [26]Al decay previously documented in very large basins (Wittmann et al., 2011, 2020). Here we find that the average $E_{Be}/E_{Al}$ value is lower than 1, implying >100,000 years of subsurface sediment storage, in basins as small as $1,000 - 10,000$ km$^2$ ($p = 0.02$). The influence of basin area is

apparent in the southern Appalachian mountains of the United States (Reusser et al., 2015; Table 5), where large
(>1,000 km$^2$, n = 5) basins have a lower average $D_{Be}/D_{Al}$ value (0.81 ± 0.05) than small basins (<30 km$^2$, n = 7,
$D_{Be}/D_{Al}$ = 0.91 ± 0.06, p = 0.017), despite other physical basin parameters being similar.
Extended sediment storage in basins as small as 1,000 km$^2$ is inconsistent with scaling frameworks that
relate sediment storage duration to floodplain area in meandering river systems (e.g., Lauer and Parker, 2008). This
suggests other mechanisms facilitate extended sediment burial and re-introduction into the active channel. In such
moderately sized basins, a variety of processes could explain extended sediment storage, including rivers cutting
into sand dunes containing long-buried sediments (Eccleshall, 2019; Vermeesch et al., 2010), hydrologically-
variable basins where flood events remobilize vertically-accreted floodplain deposits (Codilean et al., 2021), and
excavation of deeply-buried terrace sediments by outburst floods (Zhang et al., 2021). In source generation zones,
particularly on slowly-eroding hillslopes, deep vertical mixing can cause repeated burial on slopes, leading to
differential nuclide decay before sediments enter river systems (Makhubela et al., 2019).
Other physical basin parameters play secondary and interlinked roles in determining erosion rate
discordance (Fig 5, Table 1). Mean basin slope and elevation are positively correlated with each other and with
$E_{Be}/E_{Al}$ values, suggesting that alpine basins—which are typically steeper than lowland basins—produce fluvial
sediment that has experienced minimal storage and burial. Similarly, basin-averaged erosion rates and intermittent
river flow probability exhibit significant correlations with $E_{Be}/E_{Al}$ values and are negatively correlated to each other,
suggesting that slowly-eroding basins that regularly experience intermittent river flow are conducive to sediment
storage and burial. The influence of basin slope, elevation, and tectonic activity is observed when comparing basins
of similar areas in high-alpine Bhutan (Portenga et al., 2015) and low-lying eastern Australia (Codilean et al., 2021);
the Bhutan basins have $E_{Be}/E_{Al}$ values near 1 (0.98 ± 0.06, n = 11) while eastern Australian basins have lower
average $E_{Be}/E_{Al}$ values (0.83 ± 0.06, n = 7, p < 0.001) indicating extensive sediment storage (Table 5).
Based on cross-correlations between physical basin parameters, we conclude that sediment sourced from
large lowland basins— particularly those over 1,000 km$^2$, with low average erosion rates, low mean slopes, high
hypsometric integrals, and in post-orogenic settings— is more likely to exhibit erosion rate discordance indicative of
sediment storage and burial in source and/or transfer zones. Smaller alpine basins, particularly steeper basins with
higher average erosion rates in tectonically active regions, are more likely to produce sediment with $E_{Be}/E_{Al}$ values
that overlap with 1 (within 2 standard deviation analytical uncertainties), suggesting shorter and shallower sediment
storage (<10$^5$ years). We infer that this is because larger, more gently sloping basins in tectonically quiescent
regions offer more opportunities for extended sediment storage in floodplains.
**Table 5: $D_{Be}/D_{Al}$ regional case studies**

| *Location* | n | $E_{Be}/E_{Al}$ mean | $E_{Be}/E_{Al}$ S.D. | Mean basin elevation (m a.s.l.) | Mean basin slope (°) | Mean basin area (km$^2$) |
|---|---|---|---|---|---|---|
| *Southern Appalachians, USA (small basins; Reusser et al., 2015)* | 7 | 0.91 | 0.06 | 337 | 5.5 | 9 |

| | | | | | |
|---|---|---|---|---|---|
| *Southern Appalachians, USA (large basins; Reusser et al., 2015)\** | 5 | 0.81 | 0.05 | 281 | 7 | 6262 |
| *Bhutan alpine basins (Portenga et al., 2015)\*\** | 11 | 0.98 | 0.08 | 3373 | 49.4 | 164 |
| *Lockyer sub-basins, Eastern Australia (Codilean et al., 2021)* | 7 | 0.83 | 0.06 | 430 | 15.5 | 130 |

*One outlier with $E_{Be}/E_{Al}$ = 0.22 was removed. The low ratio of this sample was attributed to laboratory error in the*
*source publication.*
*\*\*For this comparison we removed basins larger than 1000 km² (n = 3)*

Climatological variables play only a minor role in the occurrence and magnitude of erosion rate

discordance. We found very weak correlations between $E_{Be}/E_{Al}$ mean annual precipitation, and aridity (Fig 5; Table
1). However, intermittent flow probability exhibited a significant negative correlation to $E_{Be}/E_{Al}$ values (Fig 5),
suggesting that basins with a higher probability of discontinuous flow for at least one day per year are more likely to
contain sediment with an extended history of burial. While fluvial systems that experience intermittent flow are most
common in arid and semiarid regions (Costigan et al., 2017), they exist around the world and intermittent flow
probability is correlated with a variety of hydrologic, geologic, and morphologic variables in addition to climate
regime (Messager et al., 2021; Figure 6). Therefore, we cannot confidently attribute an exclusively climatological
root for the correlation between intermittent flow probability and isotopic evidence of sediment burial.

Both low and high $E_{Be}/E_{Al}$ values can be caused by laboratory uncertainty (statistical measurement

uncertainty) and biases (inaccurate measurements) that influence measured nuclide concentrations. Critical to the
accuracy of $^{26}$Al and $^{10}$Be measurements by AMS is the quantification of total aluminum and beryllium in samples
(the stable isotopes, $^{27}$Al and $^{9}$Be which are many orders of magnitude greater in concentration that the radionuclides
$^{26}$Al and $^{10}$Be). Native beryllium at detectable levels in quartz is rare but occasionally present (e.g., Portenga et al.,
2015), and not all laboratories quantify total Be in samples. Unaccounted-for native $^{9}$Be will lower measured
$^{10}$Be/$^{9}$Be ratios, lower calculated $^{10}$Be concentrations, and increase calculated $^{26}$Al/$^{10}$Be ratios.

The presence of meteoric (atmospherically derived) $^{10}$Be, if not completely removed by the quartz

purification process (Kohl and Nishiizumi, 1992), will increase measured concentration of in situ $^{10}$Be as shown by
Corbett et al., (2021). In such cases, its presence lowers measured $^{26}$Al/$^{10}$Be ratios (Corbett et al., 2022; Moon et al.,
2018). Given sediment storage and thus extended residence and weathering times in large basins, the persistence of
weathered mafic minerals is more likely in smaller basin where sediment has less time to weather during transport.

Conversely, stable aluminum ($^{27}$Al) is ubiquitous in quartz, meaning that full retention and accurate

measurement of that isotope, typically via inductively coupled plasma optical emission spectroscopy after quartz
dissolution (ICP-OES; e.g., Corbett et al., 2016), is critical to properly quantifying the concentration of $^{26}$Al. Low
recovery of total Al before ICP-OES and presence of AlF complexes in ICP solutions results in lower than actual
$^{26}$Al/$^{10}$Be ratios (Bierman and Caffee, 2002; Corbett et al., 2016). While some scatter in the data is likely the result
of such laboratory errors, the observed systematic correlations between morphological basin parameters and $E_{Be}/E_{Al}$
values suggests that most low ratios are due to geologic, rather than laboratory, processes.

**5.2 Implications for cosmogenically-derived erosion rates and understanding landscapes**

Our analysis shows that nearly a third of all samples for which multi-nuclide measurements exist have discordance between erosion rates derived from $^{10}$Be and $^{26}$Al beyond 2σ uncertainty. Although some discordant samples may be the result of laboratory errors, most likely represent the complex history of sediment in drainage basins. Because our regression analysis shows that large, low-slope, low-erosion-rate basins are most likely to have sediment with discordant $^{10}$Be and $^{26}$Al-derived erosion rates, such complexity is best explained by extended sediment storage (>10$^5$ years) in low gradient floodplains typical of such basins – sufficient time for decay of $^{26}$Al to be reliably measurable (e.g., Wittmann et al., 2011; Wittmann and von Blanckenburg, 2016). However, we also find such discordance in basins as small as 1,000 km$^2$, demonstrating that extended sediment storage followed by re-entry into active channels occurs in a variety of fluvial settings in addition to large meandering, low-land river systems.

The impact of sediment storage on the veracity of cosmogenically-determined erosion rates is difficult to assess for several reasons. First, sediment samples are a mixture of material, meaning that every sample contains many thousands of sand grains, each of which has its own idiosyncratic history. Such mixing means that any attempt at decay correction will be inaccurate as mixing is a linear process and decay correction is not (Bierman and Steig, 1996). Second, sediment both loses nuclides (through radio decay) and gains nuclides (by production at depth, dominated by muons) while in storage. The resulting nuclide concentration is a convolution of time and depth in storage, where depth is unlikely to be constant through time. Because $^{10}$Be has a half-life of 1.4 My, it behaves similarly to a stable isotope on timescales typically of concern to geomorphologists, between 10$^5$ – 10$^6$ years. Thus, while low $E_{Be}/E_{Al}$ suggests sediment storage on these timescales, it need not imply that $^{10}$Be-derived erosion rates are biased significantly by radiodecay.

We consider $E_{Be}/E_{Al}$ in fluvial sediment samples as a window into watershed processes. Specifically, measuring multiple nuclides in sediment samples is useful to detect sediment storage. Additional field and remote sensing measurements, not now typically done alongside sampling for cosmogenic nuclides, have the potential to better elucidate the processes lowering $E_{Be}/E_{Al}$ and the interpretation of measured ratios. For example, field and remote sensing data could be used estimate both the volume and depth of sediment in storage on lowland floodplains (e.g., Dunne et al., 1998) whereas depth profiles along cut banks and in drill cores could provide quantification of nuclide concentrations in material stored in floodplains with depth (Bierman et al., 2005). Measuring cosmogenic nuclides in samples collected down drainage networks can demonstrate if nuclide activities and $^{26}$Al/$^{10}$Be ratios change with basin area and average slope (Clapp et al., 2002; Reusser et al., 2017). Together, these data will elucidate landscape behavior at a variety of scales and bring a deeper understanding of sediment routing and erosion rates throughout large drainage basis.

**6 Conclusions**

The discordance between basin-averaged erosion rates derived from in situ cosmogenic $^{10}$Be and $^{26}$Al in detrital fluvial samples provides insights into geomorphic controls on sediment sourcing and routing dynamics and a

valuable check on the assumption of minimal sediment storage that is central to the widely-used, single-nuclide
erosion rate method. We calculated the ratio between $^{10}$Be and $^{26}$Al-derived erosion rates ($E_{Be}/E_{Al}$) in a global
compilation of detrital fluvial samples with measurements from both nuclides (n = 766, of which n = 117 are new)
and found that nearly a third of samples (n = 238) exhibit erosion rate discordance as indicated by $E_{Be}/E_{Al} < 1$
(beyond the bounds of 2σ analytical uncertainties). Low $E_{Be}/E_{Al}$ values in detrital sediments are most likely the result
of $^{26}$Al decay during extended storage ($>10^5$ years) on hillslopes or in fluvial networks. Source basin area has the
greatest influence on sediment $E_{Be}/E_{Al}$ values, with basins $>1,000$ km$^2$ more likely to contain sediment with $E_{Be}/E_{Al}$
significantly less than 1. Other physical basin parameters have secondary and interlinked correlations to $E_{Be}/E_{Al}$,
allowing us to separate basins into two broad categories. Large, low-slope, lowland basins in post-orogenic settings
are more likely to produce sediment exhibiting erosion rate discordance indicative of extended sediment storage
($>10^5$ years). Smaller ($<1,000$ km$^2$), steep, alpine basins in tectonically active settings are more likely to produce
sediment exhibiting erosion rate agreement indicative of minimal sediment storage ($<10^5$ years). These results
provide global-scale insights into sediment routing system dynamics and demonstrate the utility of a multi-nuclide
approach for understanding geomorphic processes at the scale of drainage basins.

## Code and Data Availability

The supplementary information for this study, including supplementary data tables, text, and a Jupyter Notebook and Matlab script containing code for the statistical analyses and figure production, are available on public Github and Zenodo repositories that can be found with DOI: 10.5281/zenodo.13345369.

## Author Contributions

PB and LC conceptualized the study and acquired funding while CH conducted the investigation. PB and LC provided laboratory resources for cosmogenic nuclide chemical processing and LC supervised CH while he performed the chemistry procedures. MC performed the measurement of $^{26}$Al measurements and assisted with interpretation of results. CH and AC were responsible for compiling previously published nuclide measurement and performing geospatial analyses, while CH

performed the statistical analyses. CH prepared all data visualizations and prepared the original manuscript draft. CH, PB, LC, AC, and MC worked together to review and edit manuscript drafts, and all agreed on the final draft for journal submission.

## Competing Interests

The authors declare that they have no conflict of interest.

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
