# Peer review of "Global analysis of in situ cosmogenic 26Al and 10Be and inferred"

_Geochronology, 2024_

## Referee Comment (RC2)

The paper by Halsted et al. presents a very large dataset of cosmogenic nuclides ($^{10}$Be and $^{26}$Al) drawn from the literature, along with 121 new $^{26}$Al measurements. The samples correspond to riverbed sand covering a fairly wide portion of the globe. The authors propose an original indicator (although I do not clearly see its relevance at this stage of the paper) to characterize the complexity of the burial history of the sediments studied. Finally, simple yet well-supported statistical analyses allow for testing potential relationships between the burial indicator and a series of morphometric and climatic parameters.

The results indicate that (1) almost half of the samples show a complex exposure/burial history, and (2) there is a significant relationship between the catchment area and the complexity of sediment transport. These results thus support the recent (and quite similar) study by Wittmann et al. (2020), which obtained similar findings.

In my opinion, the value of this study lies in the size of the dataset (624 samples), the simplicity of the approach, and the clarity of the message.

The article is well-written, and the figures are clear and of good quality. In my view, this paper deserves to be published, with a few minor revisions. Below are some general comments, followed by more specific remarks throughout the text.

**General comments:**

- The title should be modified, as you are not using concentration ratios but denudation ratios. You should also add 'modern' fluvial sediments.

- The objectives and methods could be more clearly described right from the introduction. I believe your general objective is found in Lines 176-178: "We measure the morphometric and climatological properties of basins from which the sampled sediments derive and use a variety of statistical analyses to assess if basin properties are correlated with cosmogenic indications of such burial."

- In my opinion, you are missing an important potential control factor: the nature of the deposits in which the rivers evolve. For example, we recently demonstrated (Jautzy et al., 2024) a significant relationship between the proportion of glacial deposits (or LGM glacial cover) within the basins and the degree of cosmogenic imbalance. You could easily test this relationship using the glacial cover shapefiles from Ehlers et al. (2011) ⇒ https://booksite.elsevier.com/9780444534477/

- In your database, I don't see the 35 samples presumably associated with Wittmann (2011) in their publication. It seems that they only measured $^{10}$Be, and your study did not measure those. This point needs clarification.

- I understand that comparing denudation rate ratios allows you to eliminate spatial variations in production rates. However, as the paper currently stands, I don't fully grasp the actual benefit of using denudation rate ratios instead of concentration ratios. It's original, potentially interesting, and useful, but it would require more justification. Ideally, an introduction on the use of concentration ratios would be relevant, as this is the commonly used method to study sediment burial history. A simple linear regression between denudation rate ratios and concentration ratios (see figure below) confirms a very strong correlation between these two ratios. The use of the denudation ratio, therefore, needs to be better justified.

[Figure]

**Specific comments:**

Line 21: 'We test for correlations between such discordance and topographic metrics'
You also test for climatic metrics. It should appear in the abstract.

Line 46: 'Data by which to evaluate these assumptions are scarce.'
Il manque un mot ou une phrase de transition avant cette phrase.

Line 56: 'Measuring the concentrations and calculating ratios between multiple cosmogenic radionuclides has provided insight into sediment provenance (e.g., Cazes et al., 2020) and storage histories (e.g., Wittmann et al., 2011; Fülöp et al., 2020; Ben-Israel et al., 2022) in large river systems.'
Not only in large rivers. By the way, 'large rivers' should be defined somewhere in the paper.

Line 95: 'first using single nuclides and later paired nuclides'
Could you add references after each case (single and paired)?

Line 100: 'the ratio of $^{26}$Al to $^{10}$Be at production is ~6.8'
Add reference.

Line 127-131: 'arid, tropical and very large' ⇒ this is a weird way to distinguish different geographical settings. I suggest rephrasing the sentence.

Line 186-187: 'Although we identify basin properties that correlate with isotopic indications of burial and storage, the identification of specific processes responsible for storage and subsequent remobilization likely differs on a case-by-case basis.'
Yes, of course. I think you must develop this point in your Discussion.

Line 198: Maybe you could add our recent dataset ? (Jautzy et al., 2024)

Line 387-400: This paragraph deals with analytical biases. It is necessary, but I think it would be relevant to insert it in an additional sub-section that talks about the limitations of the study, also adding potential control factors for discordance, which have not been tested in this study. Such as, for instance, the nature of the deposits in which the rivers flow.

Line 427-441 (Conclusions): In view of your striking results (~50% of samples showing burial), I suggest that you reiterate in your conclusion not only the usefulness of the paired-nuclide approach, but also its necessity to verify the steady-state hypothesis, too often simply assumed in this kind of study.

---

## Author Comment (AC2)

**Response to review by Timothee Jautzy**

The paper by Halsted et al. presents a very large dataset of cosmogenic nuclides ([10]Be and [26]Al) drawn from the literature, along with 121 new [26]Al measurements. The samples correspond to riverbed sand covering a fairly wide portion of the globe. The authors propose an original indicator (although I do not clearly see its relevance at this stage of the paper) to characterize the complexity of the burial history of the sediments studied. Finally, simple yet well-supported statistical analyses allow for testing potential relationships between the burial indicator and a series of morphometric and climatic parameters.

The results indicate that (1) almost half of the samples show a complex exposure/burial history, and (2) there is a significant relationship between the catchment area and the complexity of sediment transport. These results thus support the recent (and quite similar) study by Wittmann et al. (2020), which obtained similar findings.

In my opinion, the value of this study lies in the size of the dataset (624 samples), the simplicity of the approach, and the clarity of the message. The article is well-written, and the figures are clear and of good quality. In my view, this paper deserves to be published, with a few minor revisions. Below are some general comments, followed by more specific remarks throughout the text.

*Response to summary: We thank Dr. Jautzy for this review, which offers helpful insight into revisions that will improve the manuscript. Our responses to individual comments are shown in italics below. We will make more clear in our revision the significant differences between our work/results and those of Wittman who studied only very large basins.*

**General comments:**

- The title should be modified, as you are not using concentration ratios but denudation ratios. You should also add 'modern' fluvial sediments.
    - *We agree and this was also recommended by another reviewer. We will change the title to be about denudation rather than concentration ratios and include 'modern' to describe the fluvial sediments*

- The objectives and methods could be more clearly described right from the introduction. I believe your general objective is found in Lines 176-178: "We measure the morphometric and climatological properties of basins from which the sampled sediments derive and use a variety of statistical analyses to assess if basin properties are correlated with cosmogenic indications of such burial."
    - *This is a good recommendation, we will modify the beginning of the introduction to more clearly state the objectives and methods of the study.*

- In my opinion, you are missing an important potential control factor: the nature of the deposits in which the rivers evolve. For example, we recently demonstrated (Jautzy et al., 2024) a significant relationship between the proportion of glacial deposits (or LGM glacial cover) within the basins and the degree of cosmogenic imbalance. You could easily test this relationship using the glacial cover shapefiles from Ehlers et al. (2011).
    - *This is a great suggestion. We will look at the relationship between glacial deposit cover and cosmogenic ratios across basins. The shapefiles provided in Ehlers et al. (2011) will make this relatively straightforward to test. The vast majority of the basins in this compilation do not have any modern glaciers, as the presence of glaciers has long been known to skew erosion rate interpretations. However, it is*

*likely that at least some of the basins featured here have glacial deposits from the LGM. We will be interested to see if the presence of these older glacial deposits influences observed nuclide ratios.*

- In your database, I don't see the 35 samples presumably associated with Wittmann (2011) in their publication. It seems that they only measured [10]Be, and your study did not measure those. This point needs clarification.
    - *We are confused by this comment. Wittmann et al. (2011) measured both 10Be and 26Al in their samples, and these samples are in the compilation featured here (their unique IDs begin with "WIT2011_"). We will double check out manuscript to make sure that the language we use does not make this confusing.*

- I understand that comparing denudation rate ratios allows you to eliminate spatial variations in production rates. However, as the paper currently stands, I don't fully grasp the actual benefit of using denudation rate ratios instead of concentration ratios. It's original, potentially interesting, and useful, but it would require more justification. Ideally, an introduction on the use of concentration ratios would be relevant, as this is the commonly used method to study sediment burial history. A simple linear regression between denudation rate ratios and concentration ratios (see figure below) confirms a very strong correlation between these two ratios. The use of the denudation ratio, therefore, needs to be better justified.
    - *We agree that a better explanation of the denudation rate ratio metric is needed. Reviewer #1 shared a similar sentiment, and so we will elaborate more on this metric and its advantages in the manuscript.*
    - *The linear regression shown here is highly informative, thank you. Although there is a very strong correlation between concentration ratios, we prefer the erosion rate ratio metric because it accounts for two known phenomena:*
        - *First, it accounts for known spatial variations in the Al/Be production rate ratio (Lifton et al., 2014; Halsted et al., 2021) that may introduce biases into our analysis. For example, a high-latitude sample may be mislabeled as having experienced no storage due to its concentration ratio of 6.6, but the high latitude production rate is closer to 7.0 (Corbett et al., 2017), so this sample most likely has experienced storage.*
        - *Second, using denudation rate ratios allows us to work around the curved nature of the constant exposure/erosion curves on a standard two-isotope plot. Because of this, we can differentiate between an Al/Be ratio of, say, 4 in a basin eroding at 1 mm/kyr and a basin with the same Al/Be ratio eroding at 1000 mm/kyr. In one case the ratio is at steady state and in the other it is plotting under the erosion island. Simply looking at Al/Be ratios, we would not be able to determine this.*

**Specific comments:**

Line 21: 'We test for correlations between such discordance and topographic metrics' You also test for climatic metrics. It should appear in the abstract.
o *Agreed, we will add a statement about testing climatic metrics in the abstract*

Line 46: 'Data by which to evaluate these assumptions are scarce.' Il manque un mot ou une phrase de transition avant cette phrase.
o *We will reword and double-check for grammar*

Line 56: 'Measuring the concentrations and calculating ratios between multiple cosmogenic radionuclides has provided insight into sediment provenance (e.g., Cazes et al., 2020) and storage histories (e.g., Wittmann et al., 2011; Fülöp et al., 2020; Ben-Israel et al., 2022) in large river systems.'
Not only in large rivers. By the way, 'large rivers' should be defined somewhere in the paper.
o *True, we will remove 'large'. The studies we cited mostly focused on river basins > 100,000 km$^2$, but you are correct that smaller basins are part of these studies as well.*

Line 95: 'first using single nuclides and later paired nuclides' Could you add references after each case (single and paired)?
o *Of course, this was an oversight on our part. We will include some of the original publications using these nuclides to understand river processes and erosion (Lal, 1991; Bierman and Steig, 1996)*

Line 100: 'the ratio of $^{26}$Al to $^{10}$Be at production is ~6.8' Add reference.
o *Agreed, we will add relevant references*

Line 127-131: 'arid, tropical and very large' $\Rightarrow$ this is a weird way to distinguish different geographical settings. I suggest rephrasing the sentence.
o *Agreed, we will rephrase to be more clear about different geographical settings*

Line 186-187: 'Although we identify basin properties that correlate with isotopic indications of burial and storage, the identification of specific processes responsible for storage and subsequent remobilization likely differs on a case-by-case basis.'
Yes, of course. I think you must develop this point in your Discussion.
o*Agreed, we will provide several examples demonstrating specific processes responsible for storage and remobilization in different geographic and climatic settings.*

Line 198: Maybe you could add our recent dataset? (Jautzy et al., 2024)
o *We will review this paper and, if appropriate, include it in our discussion about glacial deposits as suggested earlier. However, we are not inclined to add in these newly published results because their inclusion will require re-doing all analyses and re-making all figures in this manuscript, and we doubt that the data from these 22 basins would*

*change our overall conclusions based on the other 600+ basins we analyzed here. We can specify that the compilation here features all published dual-nuclide fluvial data available at the time of our manuscript submission, as it is inevitable that more data will trickle in before this manuscript is published.*

Line 387-400: This paragraph deals with analytical biases. It is necessary, but I think it would be relevant to insert it in an additional sub-section that talks about the limitations of the study, also adding potential control factors for discordance, which have not been tested in this study. Such as, for instance, the nature of the deposits in which the rivers flow.

o *Adding a subsection for study limitations is a great idea, it will also help with overall organization of the discussion section. After we add in the glacial deposit map suggested above and analyze correlations with denudation rate ratios and other metrics in our compilation, we will also include a statement about the nature of deposits in which the rivers flow if it seems appropriate.*

Line 427-441 (Conclusions): In view of your striking results (~50% of samples showing burial), I suggest that you reiterate in your conclusion not only the usefulness of the paired-nuclide approach, but also its necessity to verify the steady-state hypothesis, too often simply assumed in this kind of study.

o*Agreed, we think the implications of this study for the steady-state hypothesis is a major outcome of this study and should be stated more clearly, we will emphasize this more in the conclusion*

---

## Author Response (AR1)

**Response to Reviewers – Halsted et al. submission to Geochronology**
January 30, 2025

We thank both reviewers and the handling editor for their constructive feedback that helped improve this manuscript since its first submission. We implemented a few large changes that we discuss in the General Revisions section. In the following sections, we provide details about revisions implemented in response to specific editor and reviewer comments. The editor and reviewer comments are shown in italic text, while our responses are in bullet-pointed, plain text.

**General Revisions**

1. Since the first submission, we have added 142 additional samples to the compilation in this manuscript. Some of these data are newly published since our initial submission and were suggested by reviewers. Other samples are from older publications for which we did not previously have confident knowledge of AMS standards (and thus could not confidently normalize nuclide concentration measurements to modern standards), but for which we contacted the original study authors and obtained this information.

2. Based on both reviewer suggestions and discussions with colleagues, we switched from using erosion rate 'external' uncertainties to 'internal' uncertainties to estimate the 1-sigma analytical uncertainties for erosion rate ratios. We also decided on a higher threshold for samples to be considered distinguishably higher or lower than 1, this time considering 2-sigma analytical uncertainties (previously we only considered 1-sigma uncertainties when using 'external' uncertainties). Considering 2-sigma 'internal' uncertainties lowered the percentage of samples in this compilation that have erosion rate ratios distinguishably lower than 1 compared to the previous manuscript (30.5% compared to 44.2%).

3. We altered the erosion rate ratio categories used in figures throughout the manuscript. Instead of binning samples into categories based on their erosion rate ratio (e.g., 0.6-0.8, 0.8-1.0, 1.0-1.2, etc.), we instead created three categories: ratios indistinguishable from 1, ratios distinguishably below 1, and ratios distinguishably above 1 (considering 2-sigma analytical uncertainties). All figures showing samples binned into erosion rate ratio categories have been updated to reflect this new categorization scheme. We made this switch after discussions among the author team and with colleagues and decided that this categorization is a more useful and statistically rigorous depiction of the data.

**Response to associate editor Hella Wittmann:**

*I now have carefully evaluated your responses to the reviewer's comments. In the light of these, and my own thoughts on this work, I would like you to revise your paper accordingly. I will send it out for review again though. Behind this reasoning is how exactly your work complements and perhaps goes beyond what Wittmann et al. (2011; 2020) have done needs some major work and additions. I am delighted that the statistical analysis confirms what we have suggested previously, but please try to go beyond the statistical analysis. In this regard, I am looking forward to read about your explicit findings regarding hillslopes and floodplain storage in the revised manuscript. Some authors have developed models of characteristic length scales and transport velocities (e.g. Pizzuto et al.; Lauer et al.). Outcomes of these could be of help to you to further understand floodplain mixing and the response of cosmogenic nuclides. Evaluating channel forms and processes a bit deeper could help, too, to go beyond the statistics.*

- We added a more thorough discussion section (401 – 417) detailing how this compilation and analysis provide evidence for extended sediment storage in basins as small as 1,000 km2, far smaller than any basins featured in similar previous studies such as Wittmann et al. (2011; 2020). We discuss the implications of these results for existing sediment transport scaling frameworks and provide examples of how such storage could occur in basins in different physiological and climatological settings. In addition, we calculated basin hypsometric

integrals and incorporated them into our analysis to further evaluate the influence of channel morphology on erosion rate ratios.

*Please also try to implement the findings and data of Dr. Jautzy, as his work has some implications here regarding steady-state assumptions.*

- We added the data from Dr. Jautzy's recent paper and discussed the potential influence of glacial deposits on measured sediment nuclide ratios in our background section 2.2.2 (line 188), but we found that the presence of glacial deposits (here inferred from LGM ice extents) had only a weak correlation to erosion rate ratios (as shown in our updated Figure 5).

*Also, I can see and understand your reasoning behind using the ratio of denudation rates (i.e. erosion rates in the MS in some places, it would be great if you used the term "denudation" consistently), and not concentrations, for evaluation of complex sediment histories, but clearly state the limitations, too, of that approach. Be and Al do not behave exactly the same with regards to production rate scaling, if I remember correctly (I might be wrong here) so these effects would bias the denudation rate ratio, but not the commonly used concentration ratio.*

- We have revised the manuscript to ensure that we use only erosion rates in reference to calculations from in situ cosmogenic nuclide data. This is consistent with 30 years of publications from our laboratory and makes sense (as we describe in a new paragraph added to the paper for clarity, lines 109-118) because cosmogenic nuclides, produced predominately within a meter or two of Earth's surface, do not account for dissolution at depth, a major contributor to denudation (total mass loss) especially in low slope, large, tropical basins.
- The unequal spatial scaling of Be and Al production rates is represented well in the LSDn scaling scheme, which we use here. We have previously critically analyzed this scaling scheme against empirical data and found a close agreement between modeled and measured 26Al/10Be production ratios (Halsted et al., 2021). This is one of the reasons we use erosion rate ratios instead of concentration ratios, because the surface 26Al/10Be production ratio varies with latitude and altitude, so the globally averaged ratio of 6.75 cannot reliably be used everywhere in the world. We made a point to emphasize this in our revised manuscript from lines 251-254.

*Some of this may be masked by using mean elevation for denudation rate calculation from the Cronus calculator. Could you use a pixel-based approach here instead?*

- We switched from using mean elevations to an iterative process that estimates a single atmospheric pressure value that best matches the spatially-averaged nuclide production rates in each basin. This method provides a more accurate basin altitude scaling factor than mean elevations, while being less computationally intensive than pixel-based approaches, which would present a challenge in this large compilation and in some of the particularly large basins within. The method described here was implemented by one of the authors in the most recent iteration of the OCTOPUS database and was found to produce nearly indistinguishable altitude scaling factors as pixel-based methods (Codilean and Munack, 2024).

*Although I like the comment by Dr. Braucher on Section 5.2/ denudation rate discordance, please keep in mind that the 10Be does not give a "true" denudation rate, but is also affected by decay in case the ratio is below the surface production rate ratio.*

- This is a good point, and we ensured that our language in section 5.2 does not imply that 10Be-derived erosion rates will always be true, but rather that $^{26}Al/^{10}Be$ erosion rate discordance does not necessitate that 10Be-derived erosion rates are incorrect (lines 489-491).

*Also, as decay time scales are different between Al and Be, the resulting integration time scales for D are different, which may result in yet another form of bias when comparing the two.*

- We considered the role that erosion rate transience, and the different adjustment timescales of $^{26}Al$ and $^{10}Be$, could have in producing erosion rate discordance, but the long integration time scales for both nuclides means that such transience would only cause erosion rate discordance in the slowest-eroding terrains (<1 m/My), and even then it would have to be a substantial

change to cause enough erosion rate discordance to be distinguishable from non-discordance. As there is little data to suggest that such erosion rate transience has occurred in slowly-eroding basins, we did not include this as a plausible explanation for erosion rate discordance in this global compilation.

**Response to reviewer #1**

*The paper by Halsted et al. presents a very large dataset of cosmogenic nuclides ($^{10}$Be and $^{26}$Al) drawn from the literature, along with 121 new $^{26}$Al measurements. The samples correspond to riverbed sand covering a fairly wide portion of the globe. The authors propose an original indicator (although I do not clearly see its relevance at this stage of the paper) to characterize the complexity of the burial history of the sediments studied. Finally, simple yet well-supported statistical analyses allow for testing potential relationships between the burial indicator and a series of morphometric and climatic parameters.*

*The results indicate that (1) almost half of the samples show a complex exposure/burial history, and (2) there is a significant relationship between the catchment area and the complexity of sediment transport. These results thus support the recent (and quite similar) study by Wittmann et al. (2020), which obtained similar findings.*

*In my opinion, the value of this study lies in the size of the dataset (624 samples), the simplicity of the approach, and the clarity of the message. The article is well-written, and the figures are clear and of good quality. In my view, this paper deserves to be published, with a few minor revisions. Below are some general comments, followed by more specific remarks throughout the text.*

- Response to summary: We thank Dr. Jautzy for this review, which offers helpful insight into revisions that will improve the manuscript. Our responses to individual comments are shown in italics below.

**General comments:**

*The title should be modified, as you are not using concentration ratios but denudation ratios. You should also add 'modern' fluvial sediments.*
- We agree and have changed the title, although we are using "erosion" instead of "denudation" in keeping with the long tradition of cosmogenic erosion rate studies. We have changed the title to "Global analysis of in situ cosmogenic $^{26}$Al $^{10}$Be and inferred erosion rate ratios in modern fluvial sediments indicates widespread sediment storage and burial during transport"

*The objectives and methods could be more clearly described right from the introduction. I believe your general objective is found in Lines 176-178: "We measure the morphometric and climatological properties of basins from which the sampled sediments derive and use a variety of statistical analyses to assess if basin properties are correlated with cosmogenic indications of such burial."*
- We re-wrote the last paragraph of the introduction (lines 66-74) to more clearly describe our objectives and methods.

*In my opinion, you are missing an important potential control factor: the nature of the deposits in which the rivers evolve. For example, we recently demonstrated (Jautzy et al., 2024) a significant relationship between the proportion of glacial deposits (or LGM glacial cover) within the basins and the degree of cosmogenic imbalance. You could easily test this relationship using the glacial cover shapefiles from Ehlers et al. (2011).*
- We agree and have incorporated the proportion of both LGM glacial cover into our analyses. We view the LGM glacial cover as a proxy for proportion of cover by glacial deposits. We found only a weak correlation between LGM ice cover and erosion rate discordance ($R_s$ = 0.07) and ANOVA testing showed no significant differences in erosion rate discordance between basins with and without LGM ice (we used a threshold of 10% LGM ice cover to

qualify as having had LGM ice). These results are shown in the updated Figure 5, the results section 4.3 (lines 382-386), and in the supplementary Jupyter Notebook. We also added in proportion of cover by modern glaciers in each basin, although only ~8% of sampled basins have more than 1% modern glacial cover by area. We again found only weak correlation between modern day glacial cover and erosion rate discordance ($R_s = 0.11$), and the scarcity of basins with modern glacial cover that have been sampled for erosion rate analysis makes it difficult to compare these to the rest of the compilation.

*In your database, I don't see the 35 samples presumably associated with Wittmann (2011) in their publication. It seems that they only measured $^{10}$Be, and your study did not measure those. This point needs clarification.*

- I must admit that we are confused by this comment. Wittmann et al. (2011) measured both 10Be and 26Al in their samples, and these samples are in the compilation featured here (their unique IDs begin with "WIT2011_").

*I understand that comparing denudation rate ratios allows you to eliminate spatial variations in production rates. However, as the paper currently stands, I don't fully grasp the actual benefit of using denudation rate ratios instead of concentration ratios. It's original, potentially interesting, and useful, but it would require more justification. Ideally, an introduction on the use of concentration ratios would be relevant, as this is the commonly used method to study sediment burial history. A simple linear regression between denudation rate ratios and concentration ratios (see figure below) confirms a very strong correlation between these two ratios. The use of the denudation ratio, therefore, needs to be better justified.*

- We provided a more thorough explanation of the erosion rate ratio metric, including its advantages over concentration ratios, from lines 150 – 156. Although there is a very strong correlation between erosion rate ratios and concentration ratios, as is expected for the majority of the data, we prefer the erosion rate ratio metric because it accounts for known spatial variations in the Al/Be surface production rate ratio that are especially apparent at high altitudes and latitudes. In addition, using erosion rate ratios effectively eliminates the natural concentration ratio lowering that occurs during surface exposure in slowly-eroding terrains (as both Al *and* Be erosion rates will reflect this).

**Specific comments:**

*Line 21: 'We test for correlations between such discordance and topographic metrics' You also test for climatic metrics. It should appear in the abstract.*
- We changed this to "…such discordance *and both topographic and climatic metrics*…"

*Line 46: 'Data by which to evaluate these assumptions are scarce.' Il manque un mot ou une phrase de transition avant cette phrase.*
- We changed this to "Although erosion rates are now commonly measured, few studies have assessed the underlying assumptions of the technique and how often those assumptions are violated."

*Line 56: 'Measuring the concentrations and calculating ratios between multiple cosmogenic radionuclides has provided insight into sediment provenance (e.g., Cazes et al., 2020) and storage histories (e.g., Wittmann et al., 2011; Fülöp et al., 2020; Ben-Israel et al., 2022) in large river systems.' Not only in large rivers. By the way, 'large rivers' should be defined somewhere in the paper.*
- We removed "large" here because you are correct that such insight has been provided for rivers of all sizes. And we included specific basin area sizes elsewhere in the paper when referring to basin area categories.

*Line 95: 'first using single nuclides and later paired nuclides' Could you add references after each case (single and paired)?*
- Of course, this was an oversight on our part. We added some of the original publications using these nuclides to understand river processes and erosion

*Line 100: 'the ratio of $^{26}Al$ to $^{10}Be$ at production is ~6.8' Add reference.*
- We added Balco et al. (2008), which details the incorporation of the Al/Be production ratio in the denudation-rate calculator used here.

*Line 127-131: 'arid, tropical and very large' $\Rightarrow$ this is a weird way to distinguish different geographical settings. I suggest rephrasing the sentence.*
- We rephrased to read: "…in catchments across the world, ranging from arid to tropical climates and in small to very large basins." (this is now lines 143-149).

*Line 186-187: 'Although we identify basin properties that correlate with isotopic indications of burial and storage, the identification of specific processes responsible for storage and subsequent remobilization likely differs on a case-by-case basis.'*
*Yes, of course. I think you must develop this point in your Discussion.*
- We expanded our discussion section exploring some of these processes and settings, exploring some specific examples (lines 404 – 438).

*Line 198: Maybe you could add our recent dataset? (Jautzy et al., 2024)*
- We have added the 22 basins from this publication to the compilation

*Line 387-400: This paragraph deals with analytical biases. It is necessary, but I think it would be relevant to insert it in an additional sub-section that talks about the limitations of the study, also adding potential control factors for discordance, which have not been tested in this study. Such as, for instance, the nature of the deposits in which the rivers flow.*

- We expanded the section dealing with analytical biases (it is now lines 452 – 470), going into greater depths about possible laboratory uncertainty and biases, including limitations of the study. However, we did incorporate the percent cover of glacial deposits (as indicated by LGM ice cover maps) into our analyses, as discussed above, and so that is not part of the 'limitations' section.

*Line 427-441 (Conclusions): In view of your striking results (~50% of samples showing burial), I suggest that you reiterate in your conclusion not only the usefulness of the paired-nuclide approach, but also its necessity to verify the steady-state hypothesis, too often simply assumed in this kind of study.*

- Agreed, we emphasized this point more in discussion section 5.2 (lines 472-502) and in our conclusion (lines 504-507).

**Response to Regis Braucher:**

*The paper of Halsted et al. presents a statistical analysis of 624 samples from fluvial sediments where both $_{10}$Be and $_{26}$Al have been measured (among all samples, 121 new $_{26}$Al measurements are presented).*

*From these measurements and the determination of denudation rate for both nuclides, the authors state that when the two denudation rates are equal within uncertainties the sediment undergone a simple history and for more than 276 samples with denudation ratios below 1 the authors argue that burial must be involved.*

*This paper is well written and fairly present all calculations and tests performed on this dataset. I think it is worth being published in Geochronology providing some precisions and corrections.*

- We thank Dr. Braucher for his review, which we found constructive and will certainly improve our manuscript during revisions. In particular, the sensitivity test he conducted to compare different methods of Al blank correction was very informative and helpful.

*I think the title should be modified as the authors have only work on the denudation ratios, not on the concentration ratios as it is referred.*

- We agree and have changed the title, although we are using "erosion" instead of "denudation" in keeping with the long tradition of cosmogenic erosion rate studies. We have changed the title to "Global analysis of in situ cosmogenic $^{26}$Al $^{10}$Be and inferred erosion rate ratios in modern fluvial sediments indicates widespread sediment storage and burial during transport"

*Perhaps a nasty question; Except the dataset, how this paper differs from Wittmann et al. (2020)? It seems that the two papers have the same conclusion: in large floodplain the*

*probability to have a discordant denudation ratio between the two nuclides is greater than in rapid eroding settings with fast transport.*

- We feel that this paper represents a significant addition to the work conducted by Wittmann et al. (2020), both in the scope of the dataset (now 766 basins), the types of basins analyzed (large, small, and across varied climates), and in the results we find including the prevalence of extended sediment storage in basins as small as 1,000 km2 and the correlation of other physiological basin metrics including tectonic regime and hypsometric integral to the occurrence of extended sediment storage. We believe that these results, in particular the occurrence of low ratios in basins well below the size of those analyzed in Wittmann et al. (2020), represent a significant contribution. We agree that our results support the conclusions of Wittmann et al. (2020), and we emphasized this more in our revisions (see lines 405-407).

*In the abstract it is mentioned lines 32-33 that the denudation ratio study will bring a deeper understanding of sediment routing and whether erosion rate assumptions are violated. I did not see this in the present paper; I think that the authors should work on this to propose a paper that will complement the work of Wittmann et al. . If there is a length limitation in the manuscript for this, the introduction and the background sections can be reduced.*

- We have modified both the abstract and discussion substantially from the original manuscript and the study implications should now be more clear.

*I have a major concern regarding the newly presented data. Line 221-222 you mention that you correct the ams ratios by subtracting the blank ratio. This is not correct for 26Al. To do this the amount of 27Al in the samples must be the same as the one in the blank. This can be accepted for beryllium as the 9Be added in roughly the same for all samples including blanks. For 27Al the natural amount is highly variable as shown in the following figure presenting the 27Alvariation in your 121 Al samples. Therefore, you must consider subtracting the 26Al atoms (determined from the amount of 27Al added in the blank and the corresponding measured AMS ratio), form the to the 26Al amount in the sample.*

- We adopted his suggested blank correction method and have recalculated all new 26Al concentrations (and all erosion rates from these concentrations). Fortunately, we observe that the revised Al blank correction procedure did not change the overall conclusions of this paper, with the median change in calculated 26Al concentration using the revised blank correction method being less than 1%. The revised 26Al concentration calculations are in the updated Table S2.

*In Table 2S:*
- *Precise the amount of 27Al added to each blank and potentially to the sample (precise if sample are spiked or not). As the methodology follows Corbett et al (2016) I have considered 2.5 mg of spike: ok?)*
- *Is the 27Al measurement in the aliquot recalculated for the total dissolved mass?*
    - We used 1.5mg of spike and ensured the 27Al measurement in the aliquot is recalculated for the total dissolved mass as part of the calculation of 26Al atoms. These calculations are in Table S2

- *Some blank ratios are missing (see the excel file, all red sheets are the modified ones and red cells the problematic ones.*
- *Some original batch ID have different UVM Original batch number*
- *Batch 656 (CH-07) is present in the sample sheet, not in the blank one. Please harmonize these numbers.*
  - This was a mistake on our end. We found the Batch 656 blank data and have added this into the sheet.

- *Therefore, the corresponding blank is not easy to find.*

| SAP15 | 408 | CH-16 | ?? |
|---|---|---|---|
| *SAP17* | *408* | *CH-13* | |
| | | | same |
| BLK | 408 | CH-13 | |

- It seems that our organization and reporting of these newly analyzed samples led to some confusion. To try and avoid further confusion for reviewers or readers, we added extra information to the "Read Me" tab in Table S2 to specify the following:

  The "UVM Original Batch" number is the chemical processing batch in which these samples were originally run following the methods of Corbett et al. (2016). These processing batches occurred at the University of Vermont between 2009 – 2019. During those processing batches, Be and Al were separated and precipitated as hydroxide gels. The Al gels were archived from the original batches, while the Be gels continued through the remainder of the chemical processing method, eventually having 10Be measured via AMS and published. All blanks reported here are the blanks associated with the Al fractions that were extracted during this original chemical processing and stored in gel form alongside the archived Al gels.

  The "UVM Project Batch" (those beginning with the prefix "CH") refers to the re-processing batch wherein we took the archived Al gels and finished their chemical processing before sending them to the AMS for measurement (between 2019 and 2021). Because we sampled from the Al gel archive iteratively, we sometimes ran multiple samples from the same original processing batch in different re-processing batches after receiving preliminary data. So, in the example given above, the Batch 408 blank comes from the original processing batch 408, in which the Al gels for samples SAP15 and SAP17 were extracted. We reprocessed sample SAP17 in batch CH-13 alongside its original batch blank. After getting AMS data from batch CH-13, we decided to measure another SAP sample (SAP15) and included it in batch CH-16. However, we had already measured the original blank for batch 408, so this was not included in re-processing batch CH-16.

  Not all of the original processing batches had archived blanks alongside the Al gels. This is why there are no blanks associated with the original batches 401, 451, 455, 490, 497, 524, 531, 532, 533, 541, 557, 559, and 610. The absence of the blanks is probably due to

the original study PIs going back to re-sample the Al gels, as we did here, at some point over the past decade and reconstituting the blanks as well. The absence of blanks from these original batches is one of the primary reasons why we originally decided to use a project-average blank value to correct all of our Al measurements. In light of us now using the suggested batch-specific blank correction, we will use the project-average blank value to correct just these samples with no batch-specific blank.

Finally, In Table S2 Dr. Braucher pointed out that project batch CH-05 has a blank but no samples associated with it. This is because batch CH-05 contained bedrock samples during the early stages of this project, but these were removed as the project evolved into a fluvial-focused study. We mistakenly kept the blanks associated with samples in CH-05 in Table S2, even though there are no samples from this batch in the study. We will remove this blank.

*Line 231: as you only compare cosmogenic data why do you add the production rate uncertainties?*
- You are correct and we have switched to using the analytical (internal) uncertainties only in our analyses.

*Regarding the statistical analyses, I think you should move the "Morphometric and Climatological Basin Parameters – Detailed sources and procedures". From the supplement to the main text as you are using many databased from different authors.*
- We decided to keep this in the supplement because after revisions our manuscript has become quite long.

*Figure 4: for erosion rate and basin area, adjust the x-axis (crop after 2000 m/myr and after 2.5x106 km2)*
- This was a good suggestion that we implemented, cropping the basin area plot even more to $1x10^4$ km2

*In supplement add the advantage of the tests you used (why Spearman's Rank correlation, etc...); this will help.*
- We chose the Spearman's Rank correlation coefficient (rather than Pearson's) because it is a non-parametric test and the distributions of our basin parameters were mostly non-normal, we make sure to explain this in the methods section 3.5 (lines 283-284). We also made sure to clearly explain why and how other tests, such as the Tukey multiple comparison of means test, are used in this same section. We made sure to include references for each of these tests, and we feel that these references are a good source of information on these tests for the interested reader.

*Figure 6: Explain how you determined the outliers and try to mention the number of data selected per category (in the supp file)*
- We added an explanation about outlier determination in the figure caption (lines 352-353). The number of data selected per category is given in various places in section 4.3, including Tables 2 and 3 and from lines 378 – 386.

*Line 379: are you sure that mean annual precipitation and aridity are presented in Table 1?*

- Table 1 shows the results of the forward stepwise regression analysis, so only basin parameters that made a significant improvement on a linear model predicting erosion rate ratios were incorporated into the model. MAP and aridity did not improve the model and were thus not included in Table 1.

*Section 5.2 : Here you can try to develop more how the denudation ration discordance may help. From this section one can only keep in mind that the "true" denudation rate may be given by 10Be (105 – 106 years) and the denudation ratio (or the concentrations ratio, using a "banana plot") discordance can be used to show potential sediment sequestration implying a decay in 26Al concentrations.*

- We added additional discussion points to section 5.2 to develop more the utility of the erosion rate ratio metric. We explain how a ratio distinguishably lower than 1 suggests extended sediment storage ($>10^5$ years) and thus $^{26}$Al decay (lines 476-478), but how such storage does not necessarily imply that $^{10}$Be-derived erosion rates are biased significantly (lines 489-491). We expanded the last paragraph in this section (lines 492-502) to explain in more detail how this metric can also be used to elucidate watershed processes in concert with additional field and remote sensing measurements.

*Reference : Wittmann et al (2020) is mentioned twice.*

- Good catch, we will remove the repeated reference

---

## Editor Decision (ED1)

**Dear Dr. Halsted,**

apologies for my late response. I now have carefully evaluated your responses to the reviewer's comments. In the light of these, and my own thoughts on this work, I would like you to revise your paper accordingly. I will send it out for review again though. Behind this reasoning is how exactly your work complements and perhaps goes beyond what Wittmann et al. (2011; 2020) have done needs some major work and additions. I am delighted that the statistical analysis confirms what we have suggested previously, but please try to go beyond the statistical analysis. In this regard, I am looking forward to read about your explicit findings regarding hillslopes and floodplain storage in the revised manuscript. Some authors have developed models of characteristic length scales and transport velocities (e.g. Pizzuto et al.; Lauer et al.). Outcomes of these could be of help to you to further understand floodplain mixing and the response of cosmogenic nuclides. Evaluating channel forms and processes a bit deeper could help, too, to go beyond the statistics. Please also try to implement the findings and data of Dr. Jautzy, as his work has some implications here regarding steady-state assumptions. Also, I can see and understand your reasoning behind using the ratio of denudation rates (i.e. erosion rates in the MS in some places, it would be great if you used the term "denudation" consistently), and not concentrations, for evaluation of complex sediment histories, but clearly state the limitations, too, of that approach. Be and Al do not behave exactly the same with regards to production rate scaling, if I remember correctly (I might be wrong here) so these effects would bias the denudation rate ratio, but not the commonly used concentration ratio. Some of this may be masked by using mean elevation for denudation rate calculation from the Cronus calculator. Could you use a pixel-based approach here instead? Although I like the comment by Dr. Braucher on Section 5.2/ denudation rate discordance, please keep in mind that the 10Be does not give a "true" denudation rate, but is also affected by decay in case the ratio is below the surface production rate ratio. Also, as decay time scales are different between Al and Be, the resulting integration time scales for D are different, which may result in yet another form of bias when comparing the two. Many thanks for your submission to GChron. With best regards, Hella Wittmann.